# Biomacromolecules enabled dendrite-free lithium metal battery and its origin revealed by cryo-electron microscopy

Zhijin Ju[1,4], Jianwei Nai [1,4], Yao Wang[1,4], Tiefeng Liu [1,4], Jianhui Zheng[1], Huadong Yuan[1], Ouwei Sheng[1], Chengbin Jin[1], Wenkui Zhang[1], Zhong Jin [2], He Tian [3], Yujing Liu [1]* & Xinyong Tao[1]*

Metallic lithium anodes are highly promising for revolutionizing current rechargeable batteries because of their ultrahigh energy density. However, the application of lithium metal batteries is considerably impeded by lithium dendrite growth. Here, a biomacromolecule matrix obtained from the natural membrane of eggshell is introduced to control lithium growth and the mechanism is motivated by how living organisms regulate the orientation of inorganic crystals in biomineralization. Specifically, cryo-electron microscopy is utilized to probe the structure of lithium at the atomic level. The dendrites growing along the preferred < 111 > crystallographic orientation are greatly suppressed in the presence of the biomacromolecule. Furthermore, the naturally soluble chemical species in the biomacromolecules can participate in the formation of solid electrolyte interphase upon cycling, thus effectively homogenizing the lithium deposition. The lithium anodes employing bioinspired design exhibit enhanced cycling capability. This work sheds light on identifying substantial challenges in lithium anodes for developing advanced batteries.

[1] College of Materials Science and Engineering, Zhejiang University of Technology, Hangzhou 310014, China. [2] Key Laboratory of Mesoscopic Chemistry of MOE, School of Chemistry and Chemical Engineering, Nanjing University, Nanjing 210093, China. [3] Center of Electron Microscope, State Key Laboratory of Silicon Materials, School of Materials Science and Engineering, Zhejiang University, Hangzhou 310027, China. [4] These authors contributed equally: Zhijin Ju, Jianwei Nai, Yao Wang, Tiefeng Liu. *email: yujingliu@zjut.edu.cn; tao@zjut.edu.cn

Metallic lithium (Li) has been hailed the "Holy Grail" anode for Li-metal batteries (LMBs) due to its highest specific capacity (3680 mAh g$^{-1}$ or 2061 mAh cm$^{-3}$) and lowest electrochemical potential ($-3.04$ V vs. the standard hydrogen electrode)[1–3]. LMBs utilizing various cathodes, such as lithium transition-metal oxide, sulfur, and air, can reach specific energies of ~440, ~650, and ~950 Wh kg$^{-1}$, respectively, which are much higher than those of the state-of-the-art lithium-ion batteries (~250 Wh kg$^{-1}$)[4]. However, the commercialization of LMBs is hindered by the uncontrolled growth of Li dendrites and consequent low Coulombic efficiency (CE), poor cycling capability, and even safety hazards after prolonged battery operations[5–7].

Recently, a larger and growing number of researchers have been devoted to tackling the inherent issues of Li to achieve the potential application of LMBs[8–13]. It has been proven that a high local current density will result in the proliferation of dendrites[14]. Fabrication of a lithiophilic matrix is the first strategy to affect the nucleation and growth of Li and suppress dendrite formation[15–19]. Moreover, the concentration gradient of lithium ions in the electrolyte contributes to the growth of Li dendrites. Introducing a lithiophilic buffer layer between electrodes was the second promising strategy to inhibit dendrite formation and achieve good electrochemical performance[20]. For example, a polyethyleneimine sponge host[21] and glass fibre[22] were used to reduce the concentration polarization to ensure dendrite inhibition and uniform deposition of Li. In addition, constructing an artificial solid electrolyte interphase (SEI) is the third strategy to achieve stable dendrite-free Li anodes. Adding various additives to the electrolyte is one such typical method to enhance the SEI of Li anodes[23–27]. An artificial SEI with favourable mechanical properties can suppress dendrites and prevent cracking of the SEI during cycling[28–30]. Although many important developments have been made for the improvement of Li anodes in the past decade, a deeper understanding and the corresponding regulation of dendritic growth from the point of view of crystallography are highly desired. However, this area is less reported due to the challenge of characterization of crystalline Li.

For the uniform nucleation and growth of crystals, nature is undoubtedly our best mentor. Biomineralization is a widespread phenomenon in nature leading to the formation of hierarchically structured minerals by living organisms[31–33]. It has been revealed that biomolecules, particularly the stereochemical side chains within them, can guide the nucleation and growth of mineral crystals[34–37]. As a well-known example in biomineralization, the eggshell has attracted much attention, because its calcified crystallization (Fig. 1a, b) is regulated kinetically and thermodynamically by a membrane protein, resulting in uniform nucleation and oriented growth of CaCO$_3$ on the eggshell membrane (ESM) (Fig. 1c–e)[38,39]. This natural strategy is generally deemed to involve engineering the specific geometric connections between the biomolecule and a particular crystal face[40,41]. Inspired by these biomineralization phenomena, we used trifluoroethanol (TFEA)-modified ESM (TESM) to control the nucleation and growth of Li (Fig. 1f). ESM has many applications for energy storage, such as serving as a source of carbon[42,43], the precursor of synthetic separators[44], and the nanomaterial-loading substrate[45,46].

In this work, the employed TESM can effectively homogenize the lithium-ion flux to realize dendrite-free Li anodes and its regulatory mechanism is revealed by atomic-resolution cryo-transmission electron microscopy (cryo-TEM)[1,47,48]. It is found that the TESM can block the Li dendrite growth along the preferred direction and can also be incorporated inside SEI, regulating the Li deposition consequently. Li anodes modified with TESM deliver improved electrochemical properties. We believe that our findings in this work will undoubtedly bring some inspirations for the development of advanced LMBs.

## Results

**Designed biomacromolecule matrix for Li anodes.** Initially, pristine ESM was obtained by etching CaCO$_3$ in eggshells with dilute acetic acid solution. Clean and flexible ESM (Fig. 1c) can be finally obtained by a freeze-drying technique. Scanning electron microscopy (SEM) images (Fig. 1d and Supplementary Figs. 1 and 2) reveal that the ESM possesses a cross-linked three-dimensional (3D) porous structure with a nearly 90 μm thickness, which consists of interwoven and jointed protein fibres ranging from 0.5 to 2 μm in diameter. Figure 1e is the SEM image of the cross-sectional eggshell and the inset illustrates the boundary between the ESM matrix and the CaCO$_3$ shell. Given the abundant deposition sites within the ESM during calcification, the eggshell can gradually form an ordered tissue[49]. The elemental mappings for single protein fibres (Supplementary Fig. 1d) indicate the uniform distribution of C, N, O, and S within the ESM fibres. In addition, SEM images also demonstrate that many interconnected pores exist in the ESM matrix, which is favourable for the rapid diffusion of lithium ions.

As natural ESM with inevitable defects is not suitable for battery systems, we tried to modify it by one-step TFEA solvothermal treatment to reduce the possible side reactions. Due to its low dielectric constant, TFEA can stabilize the natural structure of proteins, enhance electrostatic interactions, and particularly promote α-helix structure formation in peptides[50–53]. Attenuated total reflectance Fourier transformation infrared spectroscopy (ATR-FTIR) shows a blueshift of the characteristic bands for both amide I and amide II (Supplementary Fig. 3) after TFEA modification. This results from the transformation from a β-sheet (intermolecular hydrogen bonds) to an α-helix (intramolecular hydrogen bonds) (Fig. 2a) under the very low dielectric conditions induced by TFEA[51,53,54].

The load-displacement curves for nanoindentation (Supplementary Fig. 4) show that the modulus of TESM (~200 MPa) is higher than that of ESM (~100 MPa). In addition, as shown in Fig. 2b, the lithium-ion conductivity of TESM (4.9 mS cm$^{-1}$) is higher than that of ESM (2.1 mS cm$^{-1}$). Contact angle tests (Supplementary Fig. 5) suggest that TESM possesses a remarkable affinity with the electrolytes with a contact angle of almost 0°, whereas the polypropylene (PP) separator (~44.2°) presents poor wettability with ether-based electrolytes. Moreover, the heat deterioration test (Supplementary Fig. 6) demonstrates that the TESM layer has better thermal stability and no obvious change in size and morphology at 200 °C, suggesting its unique superiority compared with the PP separator.

SEM was employed to detect the deposition behaviour of Li within the coin cells with or without TESM (Fig. 2c–f). A dendrite-free Li morphology with a spherical microstructure can be observed in the presence of TESM (TESM was coated on Li to serve as an interfacial layer), whereas long Li dendrites are everywhere on the bare Cu without the protection of TESM (Fig. 2e, f). Density functional theory (DFT) calculations were then performed to gain insights into the interaction between lithium ions and TESM. The binding energy between lithium ions and three crucial species within the peptide chains of TESM, including the peptide bonds ($E_{b1}$), the carboxyl group ($E_{b2}$), and the amino group ($E_{b3}$), were investigated (Fig. 2g). The computed results show that the binding energies of these three species (0.74, 0.38, and 0.55 eV for $E_{b1}$, $E_{b2}$, and $E_{b3}$, respectively) are all high enough to demonstrate strong affinity. In contrast, the calculated binding energy for the PP separator ($-0.29$ eV) implies a repulsive interaction, i.e., poor affinity with

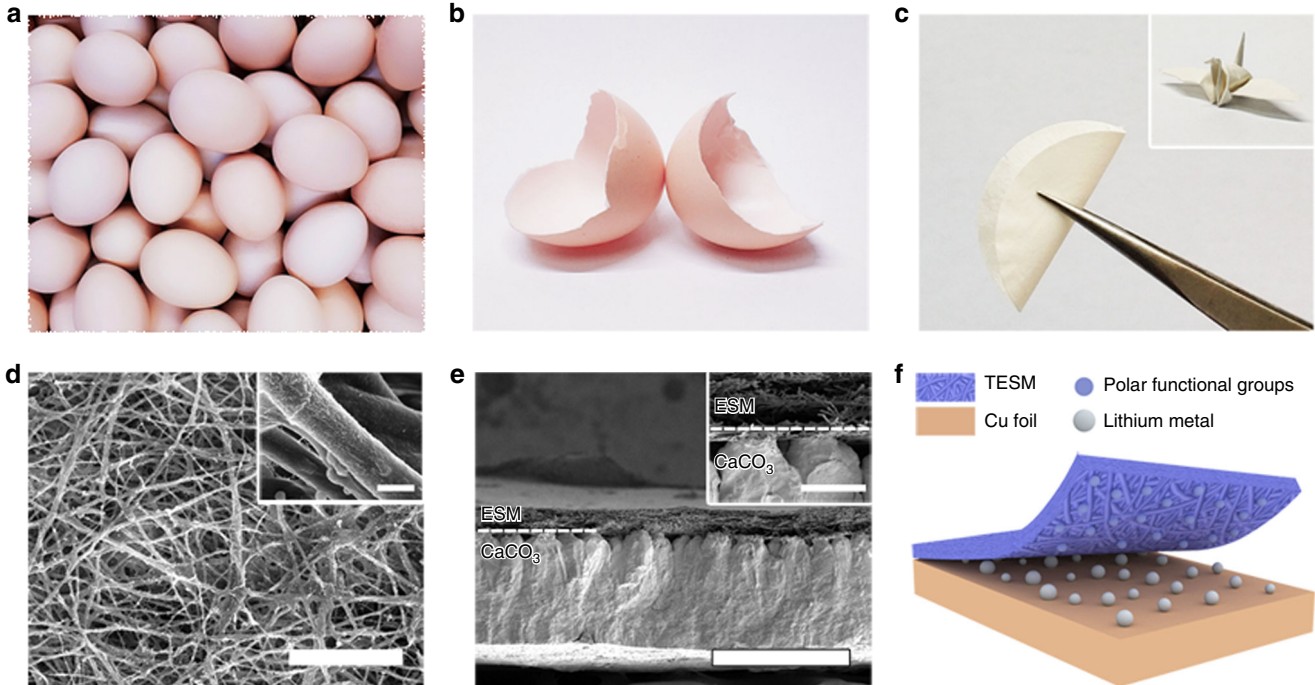

**Fig. 1 Schematic diagram of the biomacromolecule matrix employed for Li anodes. a** Digital photo of eggs. **b** Photograph of the collected eggshells. **c** The bent ESM and corresponding paper cranes folded with it (inset). **d** SEM images of the ESM and enlarged single protein fibre. Scale bars, (**d**) 50 μm and inset 1 μm. **e** SEM images of the cross-sectional eggshell and the obvious boundary between the ESM and CaCO₃ shell (inset). Scale bars, (**e**) 400 μm and inset 50 μm. **f** Li growth on TESM-modified Cu foil. The rich polar functional groups within TESM can regulate the distribution of lithium ions and realize uniform Li deposition.

the lithium ions (Supplementary Fig. 7). Moreover, inductively coupled plasma–mass spectrometry (ICP-MS) shows that the adsorption amount of lithium ions by TESM increases with the soaking time, indicating the good lithium-ion capture ability of the TESM. Therefore, with a uniform porous structure and the strong chemical affinity between lithium ions and the organic matrix, TESM may be more favourable for homogenizing the distribution of lithium ions, reducing the inner resistance of cells and inhibiting dendritic Li growth[55].

**Effects of TESM on Li nucleation and growth.** In general, the Li nuclei generated during the initial nucleation process will be easily damaged using standard TEM under constant electron-beam irradiation (Supplementary Fig. 8). Here we employed the cryo-TEM technique and found that Li metal was stable under cryogenic conditions and maintained its native state after continuous electron-beam irradiation for a few minutes. This makes understanding Li growth behaviour by characterizing its atomic structure feasible. The uniform spherical Li nuclei in the presence of TESM are observed after the first deposition in ether electrolytes (Fig. 3a, d and Supplementary Fig. 9a), which is consistent with the SEM results (Fig. 2c, d). In sharp contrast, a large amount of Li dendrites formed on the TESM-free Cu grid (Fig. 3g–i and Supplementary Fig. 9b), suggesting the effects of TESM in eliminating Li dendrites. The high-resolution TEM (HRTEM) image of the Li microsphere (Fig. 3a) from the [001] zone axis direction (Fig. 3b) exhibits a lattice spacing of 2.48 Å, which matches well with the {110} planes of Li (Fig. 3c). Li also nucleates from the {211} planes where the lattice spacing is 1.44 Å, as shown in Fig. 3e, f. The above atomic observations reveal that these Li microspheres grow preferentially along the <110> and <211> directions. In comparison, the growth direction of Li dendrites formed without the protection of TESM is either

the <110>, <211>, or <111> direction, as proved by the selected area electron diffraction patterns (insets in Fig. 3g–i). The observed growth orientations of Li dendrites here are exactly the same as those reported in a previous work where the <111> direction is preferred[48]. However, in this case, we hardly find that the deposited Li assisted by TESM grows along the <111> direction. In particular, Li deposition in standard carbonate-based electrolytes was also investigated, aiming at wide application in the future[56]. Similarly, Li nucleates with a uniform spherical morphology on the TESM-protected Cu, whereas those deposits on the bare Cu tended to be dendritic (Supplementary Fig. 10). Furthermore, we proceed to investigate how the SEI influences the nucleation and growth of Li, which is important for revealing the Li deposition behaviour[2,57,58]. According to the analysis with X-ray photoelectron spectroscopy (XPS), cryo-TEM, and other tests, the contents of N and S in Li deposited on TESM/Cu increased dramatically compared with those on bare Cu (Supplementary Figs. 11–20). These results indicate that the TESM might participate in the formation of the SEI. To verify the existence of TESM in the SEI, the bicinchoninic acid (BCA) assay was employed to detect the content of protein and the typical chemical species in TESM in the SEI of both TESM/Li and bare Li. The BCA assay explicitly demonstrated that the SEI of TESM/Li involved the TESM protein, whereas the SEI of bare Li did not (Supplementary Fig. 12), as reconfirmed by the FTIR analysis (Supplementary Fig. 13). The experiments above suggest that part of the protein, the chemical species in the TESM, is soluble in the electrolyte, which could be incorporated inside the SEI.

Apart from the intrinsically soluble property of TESM in the electrolyte, the chemical nature of TESM during the electrochemical process in terms of composing the SEI was further investigated. As seen from the CV curves, the TESM exhibited a deep cathodic current peak at ~1.4 V, whereas the bare Cu did not (Supplementary Fig. 14). Thus, this reduction peak demonstrated

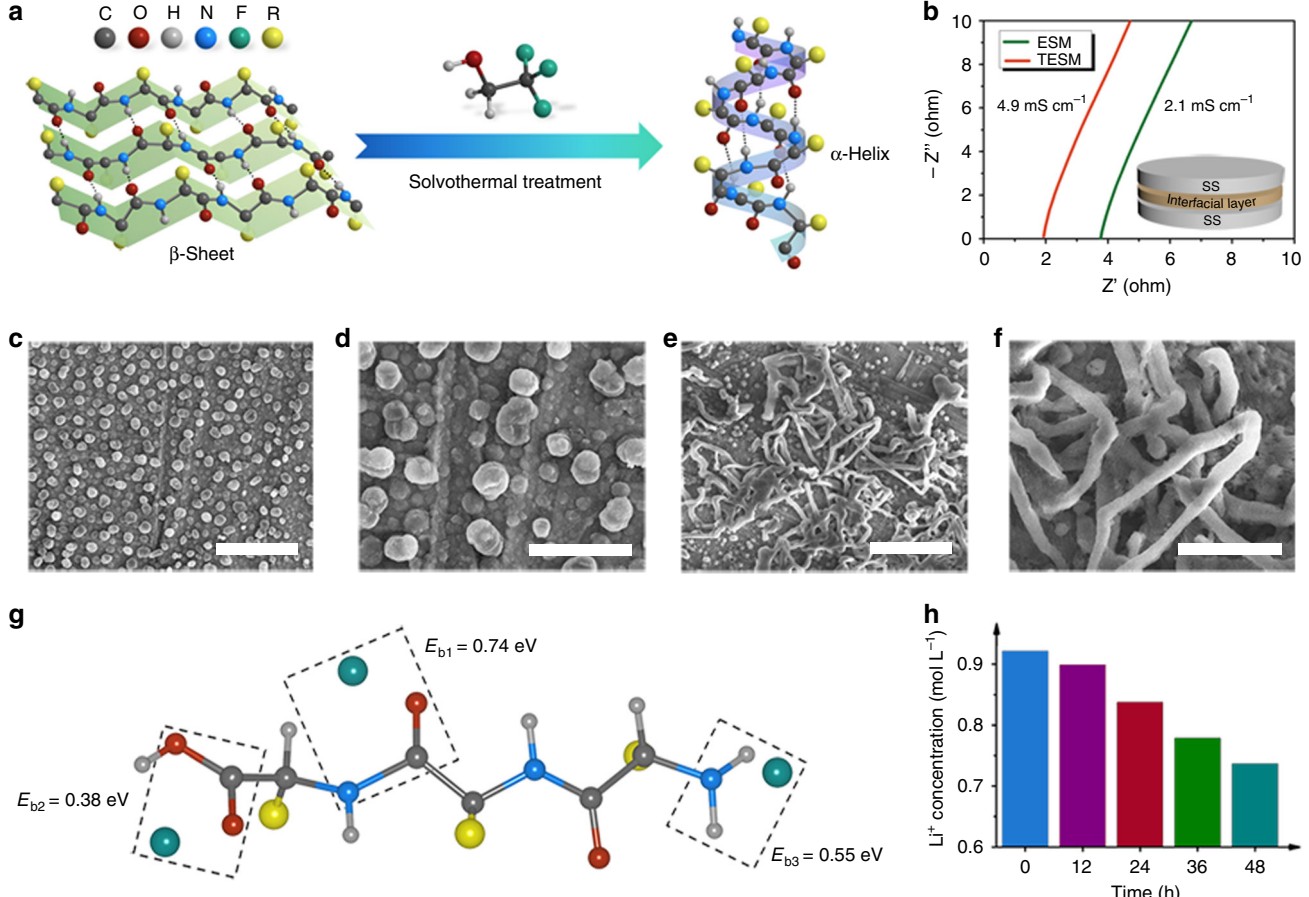

**Fig. 2 Modification of the prepared TESM and its impact on the Li anode. a** The conformational change of proteins and peptides from the β-sheet to α-helical structures induced by TFEA. **b** The ionic conductivities of ESM and TESM measured through impedance spectroscopy by assembling a cell of stainless steel/biomacromolecule matrix/stainless steel. **c–f** Comparison of SEM images of Li deposition on Cu electrodes in ether-based electrolytes with (**c, d**) and without (**e, f**) TESM. Scale bars, (**c, e**) 50 μm and (**d, f**) 20 μm. **g** The optimized structure showing the strong interactions between lithium ions and polar groups within TESM. **h** The residual lithium ions in ether-based electrolyte after soaking TESM.

that the TESM could be reduced as the frontier reaction. Although the TESM was insulated by a two-sided PP separator from the electrodes, the peaks remained close, reconfirming that the TESM is naturally soluble in the electrolyte even without lithiation. Furthermore, the cryo-TEM visualization of the SEI constructed under different potentials helped monitor the composition change (Supplementary Figs. 15–20). The mass ratio of elements N to C in the TESM present in the SEI increased significantly at −0.05 V (Supplementary Fig. 21). Considering the typical element in TESM is N, more species of TESM will form the TESM-involved SEI, particularly at −0.05 V. The TESM macromolecules embedded in the SEI might contribute to providing a homogeneous lithium-ion flux, resulting in the spherical morphology of the Li deposition.

**Electrochemical performance of TESM-modified Li anodes.** The critical role of TESM in the termination of Li dendrite proliferation was unveiled through a simple experiment. Both Li covered with TESM and bare Li were employed to investigate the difference upon cycling. The TESM-protected coin cell is silvery white (Fig. 4a), whereas the corresponding unprotected one is black (Fig. 4d). Specifically speaking, there pronounced mossy Li structures formed on the unprotected coin cell (Fig. 4e, f). In contrast, the TESM-protected Li anode has a fairly smooth and dense surface without the formation of dendritic or dead Li

(Fig. 4b, c). Even if the areal capacity increases to 3 mAh cm$^{-2}$, TESM/Li still possesses a rather uniform morphology compared with that of the bare Li after the cyclic operation (Supplementary Fig. 22).

The electrochemical effect of TESM on Li was further evaluated by identifying the cyclability and CE of the Li anode. As displayed in Fig. 4g, all the cells equipped with TESM exhibit reinforced stabilities at various current densities. At a low current density of 1 mA cm$^{-2}$, the TESM-modified electrode can maintain 98% over 200 cycles and 97% over 320 cycles (Fig. 4g and Supplementary Fig. 23a, respectively). Even at a higher current density of 5 mA cm$^{-2}$, the TESM-modified electrode still has an average CE of 96% over 140 cycles. Moreover, both the cyclability and CE for cells with TESM were also explored at high areal capacity (Supplementary Fig. 23b, c). Regardless of the operation at any current density and areal capacity, the cells with bare Cu suffer a terrible cycling performance and inferior CE values, consequently falling into fast decay. Such a difference in the cells with TESM modification and bare Cu clearly confirms the effective suppression of Li dendrite formation by TESM. It is important to note that cells with ESM at various current densities have lower initial CEs than the TESM-modified ones. This might be attributed to the severe irreversible reactions that stem from the defect sites of pristine proteins. In addition, to eliminate the contribution of the good wettability for TESM, an alternative glass fibre with superior wettability was used under the same

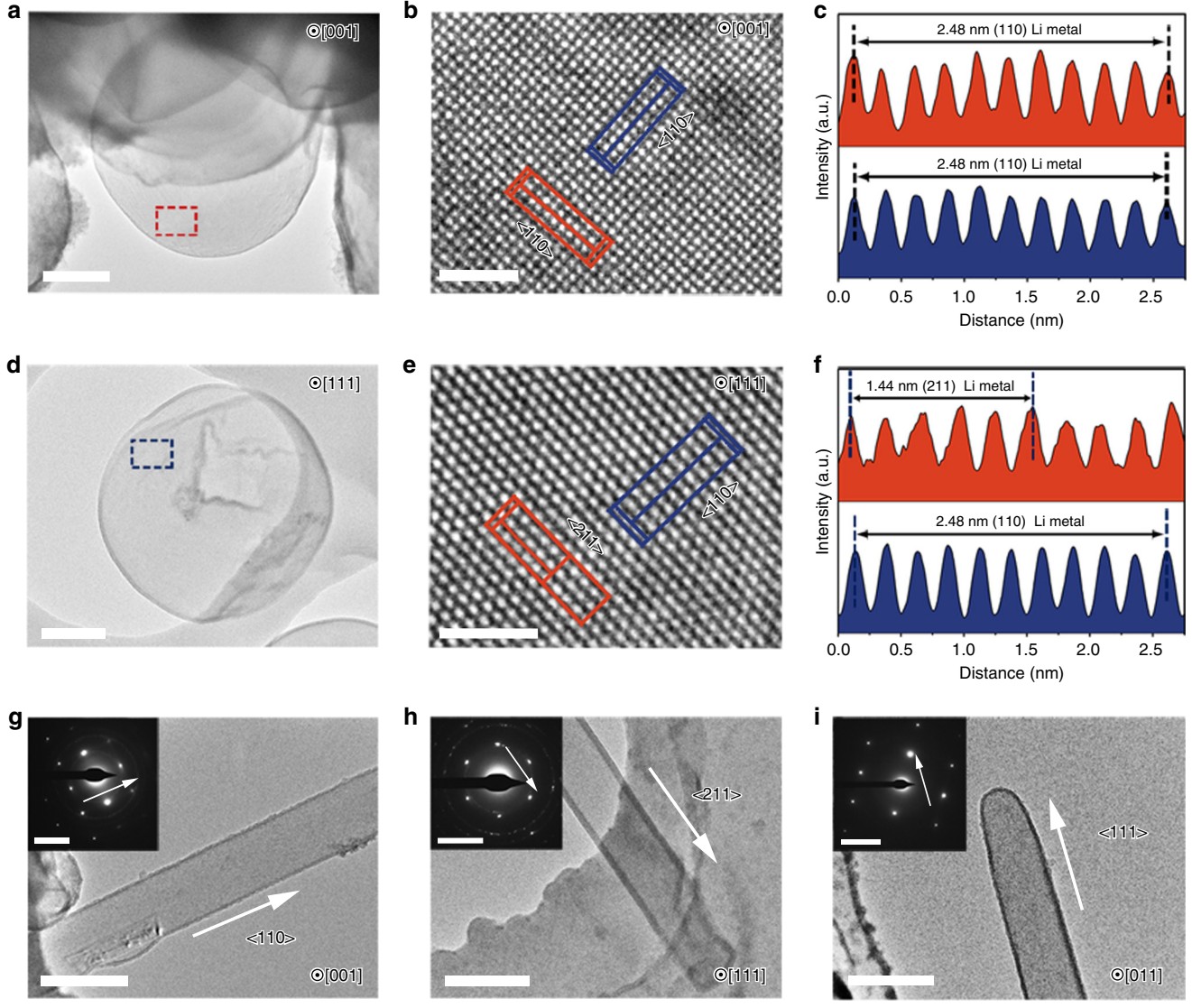

**Fig. 3 TESM-induced dendritic suppression mechanism. a**, **d** Cryo-TEM images of Li microspheres deposited in the presence of TESM in ether-based electrolytes. **b**, **e** The corresponding HRTEM images of the marked area in **a** and **d** along the [001] and [111] zone axes, respectively. **c**, **f** The measured lattice spacing in **b** and **e**, respectively. **g–i** Cryo-TEM images of Li dendrites formed in ether-based electrolytes without TESM and their corresponding SAED patterns (insets) growing along the (**g**) <110>, (**h**) <211>, and (**i**) <111> directions. Scale bars: (**a**, **d**) 500 nm, (**b**, **e**) 2 nm, (**g–i**) 500 nm, and insets 5 nm⁻¹.

conditions (Supplementary Fig. 24). Despite the improved CE, the results are still inferior to the performances of the TESM or ESM protected electrodes.

In a further evaluation, the stripping/plating profiles for electrodes at 3 mA cm⁻² are depicted in Fig. 4h and the inset shows the enlarged curves. The TESM-modified Cu, ESM-modified Cu, and bare Cu cells possess voltage hysteresis of 97, 203, and 123 mV, respectively. Notably, the difference between TESM and ESM indicates that the TFEA treatment effectively enhances the affinity for lithium ions, enabling the TESM-modified electrode to have the lowest voltage hysteresis. In addition, the function of the voltage hysteresis in cycling is described in Fig. 4i. The hysteresis of the TESM-modified cells is still the smallest among the samples. Subsequently, the morphologies of Li deposition on the Cu foil current collector after 30 cycles at 2 mA cm⁻² were surveyed as well. Compared with the large amount of dendrites on bare Cu, the homogenous Li deposition and dendrite-free morphology with TESM can be realized (Supplementary Fig. 25).

To further evaluate cycling performance, symmetric coin cells were assembled and tested under different current densities with an areal capacity of 1 mAh cm⁻². At a current density of 1 mA cm⁻², the symmetric cells of TESM/Li exhibit superior electrochemical performance in terms of a much smaller overpotential of 12 mV and ultralong stability of ~2000 h (Fig. 5a). In contrast, the ESM/Li symmetric cells, as expected, have a much higher overpotential at the corresponding current densities, which may result from inevitable side reactions upon cycling. In addition, the cells with bare Li foil suffer from a larger and gradually increasing overpotential, which is ascribed to the formation of Li dendrites along with the continuous repair of SEI fractures during the Li plating/stripping process[59].

As the current density increases to 5 mA cm⁻², cells with bare Li foil have difficulty cycling for over 180 h (Fig. 5b). There exists a sudden voltage enlargement for bare Li, indicating cell failure. In contrast, TESM-involved cells have a flat voltage plateau with a low overpotential of ~45 mV for over 3000 cycles (>1200 h). Compared with other reported works (Supplementary Table 1),

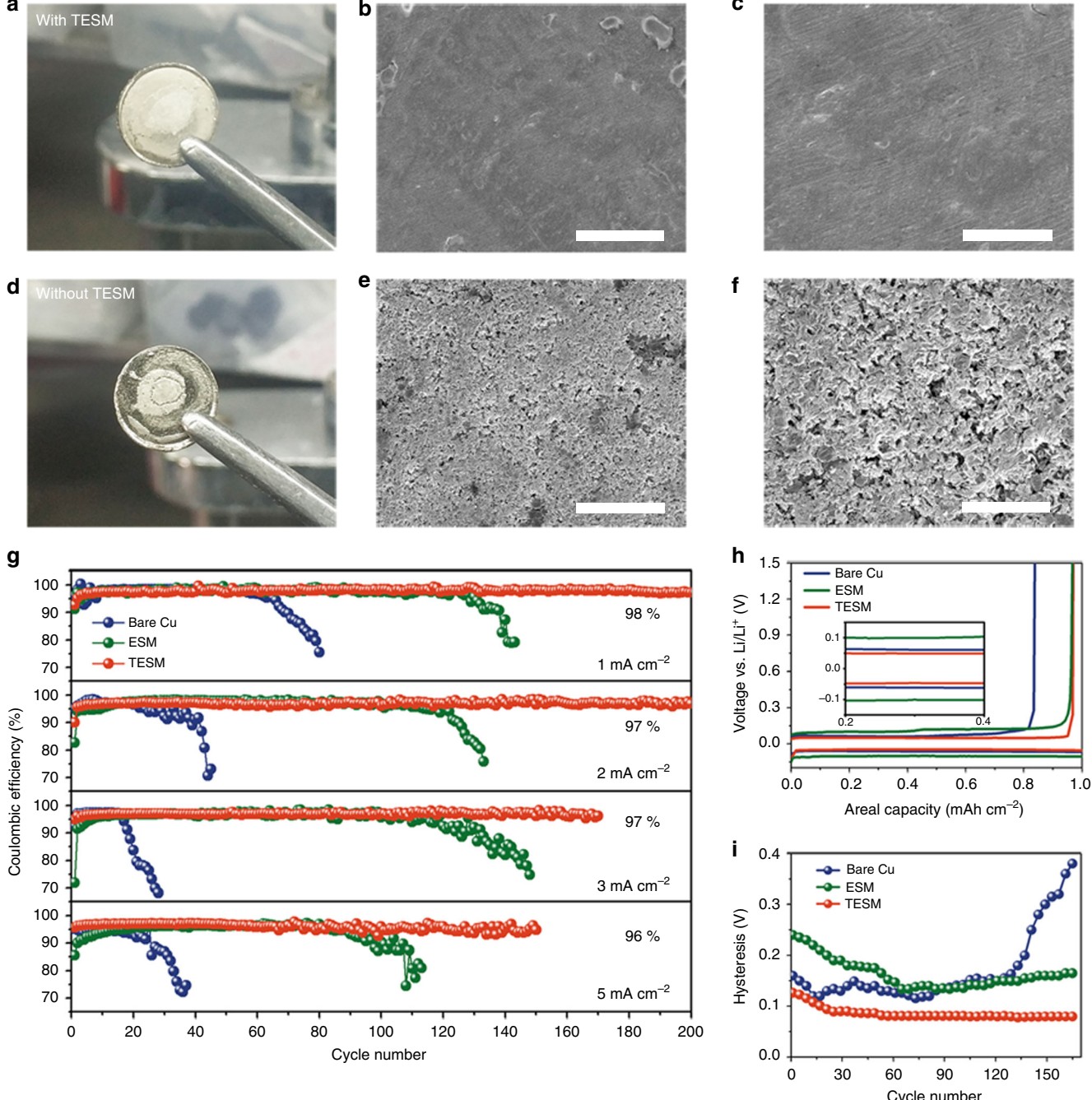

**Fig. 4 Morphology and cycling stability of a Li anode assisted by TESM in ether-based electrolytes. a, d** Optical photographs of (**a**) Li foil protected by TESM and (**d**) bare Li foil after 20 cycles at 2 mA cm$^{-2}$ for 1 mAh cm$^{-2}$. **b, c, e, f** SEM images and corresponding enlarged areas of (**b, c**) TESM-protected Li and (**e, f**) unprotected Li, respectively. Scale bars, (**b, e**) 50 μm and (**c, f**) 20 μm. **g** Comparison of CEs with bare Cu, ESM-modified Cu, and TESM-modified Cu at various current densities. The fixed cycling capacity in each cycle is 1 mAh cm$^{-2}$. **h** The voltage profiles for the electrodes at 3 mA cm$^{-2}$. **i** The corresponding voltage hysteresis of electrodes with a current density of 3 mA cm$^{-2}$.

TESM/Li has a competitive performance in terms of overpotential and lifetime. Even at high current densities of 3 or 10 mA cm$^{-2}$, similar trends in the TESM/Li symmetric cells are also readily achieved (Supplementary Fig. 26a, b). When the areal capacity increases to 2 mAh cm$^{-2}$, cells with TESM can also deliver improved cycling stability and lifetime compared with those of the bare Li or ESM/Li counterparts (Supplementary Fig. 26c). Considering the request for wide application in pairing with lithium-ion cathodes, we then improved the areal capacity to above 3 mAh cm$^{-2}$ [60]. The TESM/Li anodes under 3 and 5 mAh cm$^{-2}$ both show stable cycling performances for over 600 h with lower overpotentials of ~15 mV and ~40 mV, respectively (Fig. 5c, d). To investigate the interfacial stability, Nyquist plots for cells with bare Li foil, ESM/Li, or TESM/Li electrodes after 1 and 10 cycles were collected. Notably, TESM/Li exhibits low and stable interfacial resistance (Supplementary Fig. 27), consistent with the lower overpotential in the plating/stripping process.

The high reversibility of the TESM/Li anode encouraged us to perform a full-cell test by pairing it with a commercial LiFePO$_4$ cathode. The cycling stability enhanced by the TESM was

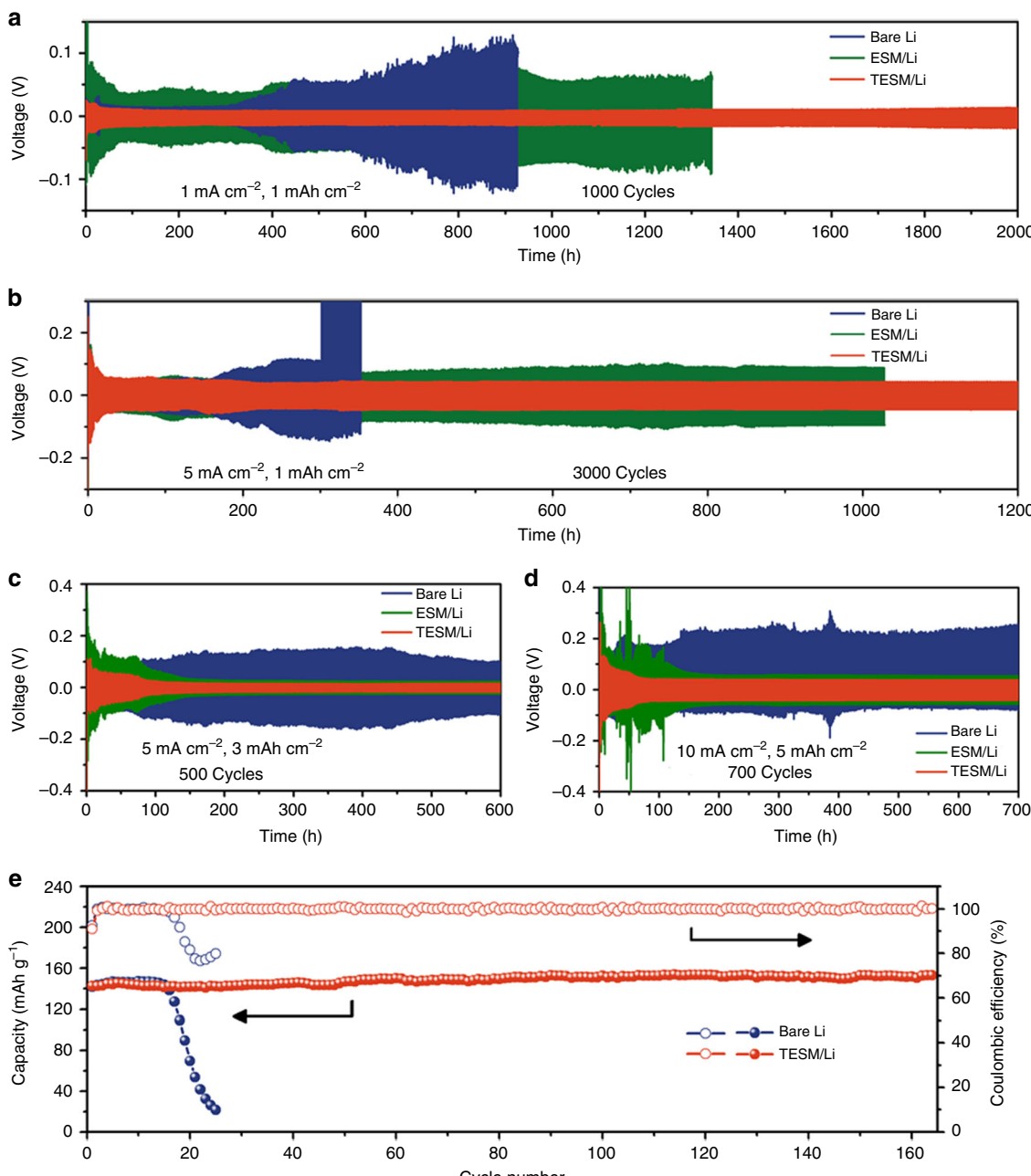

**Fig. 5 Electrochemical performance of bare Li, ESM/Li, or TESM/Li electrodes in ether-based electrolytes. a–d** Voltage profiles of symmetric cells at (**a**) 1 mA cm$^{-2}$, (**b, c**) 5 mA cm$^{-2}$, or (**d**) 10 mA cm$^{-2}$ with a Li capacity of (**a, b**) 1 mAh cm$^{-2}$, (**c**) 3 mAh cm$^{-2}$, or (**d**) 5 mAhcm$^{-2}$. **e** Cycling performance of Li/LiFePO$_4$ full cells with a capacity ratio of the negative electrode to the positive electrode (N/P ratio) of ~3.3 (Cu with limited Li: 10 mAh cm$^{-2}$, LiFePO$_4$: 3 mAh cm$^{-2}$) at 1 C (1 C = 170 mA g$^{-1}$).

evaluated under high mass loading of LiFePO$_4$ (~3.0 mAh cm$^{-2}$) and low N/P ratio (~3.3) to meet realistic conditions. As shown in Fig. 5e, the cells with TESM/Li show a prolonged lifespan of more than 160 cycles and the capacity is still above 150 mAh g$^{-1}$ with a steady CE of nearly 100%. In sharp contrast, cells with bare Li have a fast decay in capacity and cycling (below 50% capacity retention after 20 cycles). These results demonstrate that TESM is an ideal interfacial protective layer for Li anodes to suppress Li dendrites, possessing great potential for high-energy-density LMBs.

## Discussion
Nature has gifted us tremendous inspiration to design materials for addressing challenging issues. Inspired by biomineralization

through which the morphology and orientation of crystals are regulated by a functionalized organic matrix, we proposed a strategy of using a fibrous ESM with abundant lithiophilic sites to engineer Li deposits aiming at a dendrite-free Li anode. Through simple solvothermal treatment of ESM, TESM is obtained, achieving improved mechanical properties and ionic conductivity. DFT calculations and ICP-MS adsorption experiments (Fig. 2g, h) indicate that lithium ions are prone to be adsorbed onto TESM due to the strong binding interactions. TESM with a 3D network can homogenize the distribution of lithium ions to decrease the lithium-ion concentration gradient and remediate the locally enhanced lithium-ion flux, consequently restraining the formation of Li dendrites during prolonged battery operation

(Fig. 1f). In particular, we employ cryo-TEM, an emerging detection technique to atomically visualize sensitive battery materials[1,48,56,58], to reveal the fundamental mechanism for Li dendrite suppression. Li deposition in the presence of TESM generates abundant Li microspheres (Fig. 3a, d and Supplementary Fig. 9a), whereas many Li dendrites appear on bare Cu (Fig. 3g–i and Supplementary Fig. 9b). Importantly, the obtained HRTEM images clearly exhibit atom alignment and confirm that the typical dendritic growth along the <111> direction is inhibited with the assistance of TESM[51]. This conclusion is consistent with the current understanding of how living organisms create structured minerals[40,61]. In addition, the BCA protein assay and FTIR analysis confirm that some of the soluble protein species in TESM contribute to constructing the SEI (Supplementary Figs. 12 and 13). In detail, TESM is initially reduced to generate a passivation layer and to intensively participate in the SEI formation below 0 V, thus enabling homogeneous Li deposition during cycling. According to a recent report, the addition of $NO_3^-$ could generate a low-resistance SEI and thus contribute to the Li deposits on the bulk surface instead of the preferential dendritic deposition on the tips/kinks[56]. Hence, improving the migration of lithium ions across the SEI is crucial in generating uniform spherical Li deposits. Similarly, in this work, the cell with TESM/Li exhibits a dramatically lower resistance compared with that with bare Li (Supplementary Fig. 27). As such, the protein-based macromolecules embedded in the SEI might induce the nucleation and growth of Li spheres by reducing the resistance, indicating another possible reason for suppressing Li dendrites.

In conclusion, we demonstrated a biomineralization-inspired design by introducing TESM to guide dendrite-free Li deposition. Through protection by TESM, the Li anode achieves excellent cycling stability. Li electrodeposition at high current density (5 mA cm$^{-2}$), which generally results in severe dendritic growth, exhibits an ultralong cycling life of over 1200 h. Moreover, when paired with the LiFePO$_4$ cathode, full cells fabricated with the TESM/Li anode exhibit improved cyclic stability even with high loading of LiFePO$_4$ and a low N/P ratio. As such, this bioinspired strategy might be relevant for fundamentally understanding the critical factor for dendrite inhibition from the perspective of atomic resolution, towards realizing high-safety long-cycling LMBs.

## Methods
**ESM and TESM preparations**. The eggshells were first collected from common eggs purchased in a local market. After being rinsed with deionized water, the eggshells were then immersed into a 25% acetic acid solution overnight to etch the CaCO$_3$ in the outer shell. Subsequently, the light-pink ESM was obtained after separating the etched CaCO$_3$. Furthermore, the as-prepared ESM was placed into a 90% TFEA solution in an oven (~80 °C) overnight to prepare TESM. The primarily obtained TESM was purified by deionized water rinsing to eliminate the residues. Finally, the TESM was treated by a freeze-drying procedure for ~24 h.

**Characterization**. A field-emission SEM (Nova NanoSEM 450) was utilized to characterize the morphology and microstructure of ESM and TESM, as well as the Li deposits. TEM (FEI Talos-S) with a Gatan 698 cryo-transfer holder was employed for Cryo-TEM characterizations. The samples for Cryo-TEM were prepared by depositing Li on a TEM grid at 1 mA cm$^{-2}$ with a capacity of 0.5 mAh cm$^{-2}$ in ether-based electrolytes. ATR-FTIR was carried out on an infrared spectrophotometer (Nicolet 6700). For nanoindentation tests, an Agilent G200 nanoindentation system with a maximum indentation depth of 2000 nm was utilized based on the continuous stiffness method mode at room temperature. We employed an OCA50AF Contact Angle System (Dataphysics Corp., Germany) with the ether-based electrolyte to measure the contact angle. ICP-MS was conducted using PerkinElmer, Elan DRC-e. XPS measurements were conducted on a Kratos AXIS Ultra DLD spectrometer. BCA assays of the SEIs on TESM/Li and bare Li were obtained after 20 cycles at 2 mA cm$^{-2}$ with 1 mAh cm$^{-2}$ in ether-based electrolytes. To measure the absorbance, a chromogenic reagent was added into the wells of a transparent 96-well plate, followed by the indicated amount of the

protein standard and test samples (20 μL). After incubation at 37 °C for 30 min, the absorbance at 562 nm was measured on a multimode microplate reader (SpectraMax M5).

**DFT calculations**. Our calculations were based on DFT as implemented in the Vienna Ab initio Simulation Package[62,63]. The generalized gradient approximation within Perdew-Burke-Ernzerhof and projected augmented wave pseudopotentials were used to describe exchange-correlation functionals and electron-ion interactions[64,65]. We employed a kinetic energy cutoff above 500 eV for the plane-wave expansion. Geometries were optimized until the force was converged to 0.02 eV Å$^{-1}$ for all the calculations. To eliminate interactions between neighbouring images, a vacuum space of at least 15 Å was utilized for all calculations.
The binding energy $E_b$ was defined as follows:

$$E_b = E_{TESM} + E_{Li} - E_{Total} \qquad (1)$$

where $E_{TESM}$, $E_{Li}$, and $E_{Total}$ represent the total energies of the peptide chain of TESM, lithium ion, and the whole system, respectively.

**Electrochemical measurements**. An Ar-filled glove box (H$_2$O < 0.1 p.p.m., O$_2$ < 0.1 p.p.m.) was employed to assemble CR2032-type coin cells. Li foil was used as the counter electrode and Cu foil with TESM served as the modified working electrode. The half cells of Li-Cu and Li-Li were tested in ether-based electrolytes. The ether-based electrolyte was prepared by adding LiTFSI (1.0 M) in dioxolane (DOL) and dimethoxyethane (DME) (DOL/DME, 1:1 by volume) with 1 wt% LiNO$_3$ additive. The carbonate-based electrolyte was prepared by adding LiPF$_6$ (1.0 M) in ethylene carbonate (EC), diethyl carbonate (DEC), and ethyl methyl carbonate (EMC) (EC/DEC/EMC, 1:1:1 by volume) with 1 wt% fluoroethylene carbonate additive. The electrochemical performance of the coin cells was tested using a Neware multichannel battery cycler. To stabilize the SEI, three cycles were performed in the range of 0.01–1.0 V at 0.05 mA cm$^{-2}$. Subsequently, a Li capacity of 1 or 2 mAh cm$^{-2}$ was plated onto the current collector and followed by stripping Li with a cutoff voltage of 1.5 V for every cycle. We also measured Li-Li symmetric cells, which were cycled at 1, 3, 5, and 10 mA cm$^{-2}$ with Li capacities of 1, 2, 3, or 5 mAh cm$^{-2}$. The CV measurements were carried out on a CHI660 electrochemical workstation. The impedance spectroscopy spectrum was measured over a frequency range from 1 MHz to 1 mHz using a CHI660 electrochemical workstation. The ionic conductivity of interfacial protective layers was measured using the same method. The Li/LiFePO$_4$ full cells were composed of an anode of bare Li or TESM/Li, a piece of Celgard 2400 separator, and a LiFePO$_4$ cathode. The cathode was composed of 80 wt% LiFePO$_4$, 10 wt% polyvinylidene difluoride binder, and 10 wt% Super P conductive carbon black. The electrolyte volume used in this work was 60 μL, the N/P ratio[20] was ~3.3 (limited Li was deposited on the Cu current collector with a fixed capacity of 10.0 mAh cm$^{-2}$ at 1 mA cm$^{-2}$ in ether-based electrolytes), and the mass loading of the LiFePO$_4$ cathode was ~3.0 mAh cm$^{-2}$ (~22.4 mg cm$^{-2}$). The diameter of the cathode and Cu current collector was 12 mm. All Li/LiFePO$_4$ full cells were tested between 2.5 and 4.2 V at 1 C in ether-based electrolytes.

## Data availability
The data that support the findings of this study are available from the corresponding author upon request.

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

## Acknowledgements

We acknowledge financial support from the National Natural Science Foundation of China (Grant numbers 51722210, 51972285, U1802254, 11904317, and 21902144), the

Natural Science Foundation of Zhejiang Province (Grant numbers LY17E020010 and LD18E020003), and the high-performance computational resources (TianHe-2) provided by LvLiang Cloud Computing Center of China.

## Author contributions

These authors contributed equally: Z.J.J., J.W.N., Y.W., T.F.L. Z.J.J., Y.J.L. and X.Y.T. designed and conceived this study. Z.J.J., T.F.L., H.D.Y., C.B.J. and O.W.S. contributed to the materials synthesis, characterizations including cryo-TEM analysis, electrochemical measurement, and data analysis. The DFT calculations were performed by J.H.Z. and Y.W. J.W.N., Y.J.L., H.T., Z.J., W.K.Z., T.F.L. and Y.W. provided important guidance for the experiment. Z.J.J., Y.J.L. and X.Y.T. drafted the manuscript. All authors contributed to the analysis and discussed the results leading to the final version of the manuscript.

## Competing interests

The authors declare no competing interests.
