## [Peer Review File · Nature Communications]

Reviewers' comments:

Reviewer #1 (Remarks to the Author):

Summary:

In this work to improve battery performance, Tao and colleagues fabricate a membrane film composed of an eggshell protein (isolated from commercial eggshells) that is further chemically modified by a solvothermal process with trifluoroethanol. This porous membrane film termed “TESM” is then used as a coating on either a Cu or metallic Li foil substrate to influence the nucleation and growth of Li metal. The authors propose that in addition to homogenizing the distribution of Li ions near the substrate, the peptide chains in the TESHM show an increased binding energy to (111) plane of Li metal according to DFT calculations. Both theoretical (DFT) and experimental (e.g. cryo-EM) techniques are used to support this claim. Improved electrochemical performance of both full cells and half cells are demonstrated with the TESHM coating strategy.

Review:

Generally, it is quite interesting and striking that the normally dendritic Li metal grows as a spherical morphology with the TESHM film (assuming this effect is in carbonate-based electrolyte). The only other work this reviewer is aware of that shows this effect in carbonate electrolytes is (*Nature communications* 9.1 (2018): 3656.). However, this manuscript lacks detail regarding the experimental procedures and protocols necessary to fully judge both the proposed mechanism and the electrochemical performance. If the mechanism of how Li metal changes from a dendritic to spherical morphology can be explained, this work would be of interest to the battery community. Unfortunately, this is not the case for the manuscript in its current form and this reviewer cannot support publication of this work in *Nature Communications* until the issues below are appropriately addressed.

1. The cryo-EM images in Fig. 3 are indeed beautifully taken, showing the crystalline Li metal along various zone axes. However, no detail is provided as to which electrolyte the Li metal was formed in. Upon first glance, Fig. 3a and 3d appear to be grown in DOL/DME electrolyte with 1% nitrate, which is widely known to nucleate Li metal in a spherical morphology. Fig. 3g-i then appear to be grown in standard carbonate-based electrolytes. Please provide details here. This reviewer will assume the electrolyte used in Fig. 3a,d and 3g-i are both carbonate-based electrolytes, **otherwise there is no novelty as ether electrolytes with nitrate additives are known to influence the Li metal morphology.**
2. The authors claim that with the TESHM membrane, dendrite growth is suppressed due to a combination of providing a uniform distribution of Li ions and a direct influence on Li metal nucleation. There may indeed be an effect of the chemical groups of the TESHM to help reduce the concentration gradient of Li ions. However, the explanation regarding the

interaction between the peptide chains of the TESM and various crystalline facets of the Li metal does not make sense. Authors state that “binding energy of peptide chains on Li (110) and (211) slabs are lower than that on Li (111) slab” (line 160-163), which inhibits “growth of Li deposits along $\langle 111 \rangle$ direction” (line 169). If this were the case, Li metal should not grow as spherical structures and instead continue to grow as dendritic structures, just along the $\langle 211 \rangle$ or $\langle 110 \rangle$ directions. In order to grow as a spherical morphology, the Li metal would need to grow isotropically (by definition) with no preference towards any crystalline facet. However, the DFT calculations by the authors reveal otherwise. Furthermore, the calculations fail to consider the effect of the solid electrolyte interphase (SEI), which prevents direct contact between the Li metal surface and the peptide chains in the TESM. It is possible that chemical species in the TESM are soluble in electrolyte which would then directly influence the Li metal nucleation. This should then be studied using cryo-EM imaging and spectroscopy to see if such species are incorporated into the SEI.

3. In order to fairly judge the electrochemical performance of both the control and sample of interest, authors need to provide detailed information on battery components. For example, the electrolyte volume and N:P ratio, which greatly impact battery performance (*Nature nanotechnology* 14.6 (2019): 594.), need to be explicitly stated for each set of electrochemical data presented in the manuscript and supplemental materials.
4. How was the TESM placed onto the Cu or Li metal substrate? Does it serve as the separator or was the TESM a coating? Please provide details in either main text or supplemental materials.

Reviewer #2 (Remarks to the Author):

This manuscript describes a facile approach of natural biomolecules utilization for generating dendrite-free lithium metal anodes. This bio-inspired design proposes a relatively unique experiment, but cleverly performed and with interesting results: the organic matrix derived from trifluoroethanol-modified eggshell membrane significantly reduces the oriented growth of lithium dendrites along $\langle 111 \rangle$, achieving the lithium metal-based rechargeable battery with significantly boosting cyclic performance. What impresses me is the atomic-resolution imaging through cryo-transmission electron microscopy, in visualizing the origins of lithium dendrites killing. The clear exhibition of the lithium atoms alignments coupled with computational simulation confirm that the remarkable adsorptions between biomolecule groups and specific lithium crystal faces are responsible for dendritic preferred $\langle 111 \rangle$ inhibition. The mechanism above is convincing and consistent well with how lives construct hierarchical crystalline structure through biomineralization. The results presented by authors have motivated a novel insight for deeply understanding the troublesome lithium dendrite growth, regarding controlling the lithium crystallization, which is even ingeniously inspired and learned from nature.

In conclusion, I like the story, and I recommend the publication of this manuscript in Nature Communications. Before the acceptance, I have however some comments that require explanations:

1. In Line 5, Page 6, the statement "Therefore, with the enhanced mechanical property and lithium-ion conductivity, the TESM may be more favorable for homogenizing the distribution of lithium ions, reducing inner resistance of cells, and inhibiting the dendritic Li growth.", in my opinion, is not accuracy enough considering the causality of results and conclusions. To inhibit the lithium dendrites through homogenizing the lithium ions and thus reducing the local lithium ions flux should be attributed to the reason of both the uniform porous structure and the strong chemical affinity between lithium ions and organic matrix. The authors should rephrase the discussion of this part at the proper position in manuscript.
2. In DFT calculations, why do the authors construct two configurations, i.e., peptide chain with or without lithiation on Li slab, and what is the difference between these two situations since the results almost remain unchanged. This hypothesis is crucial to the conclusion so the authors should offer us more detailed discussion.
3. The enhanced cycling capabilities of full cells using TESM / lithium metal are demonstrated by making a comparison of capacity retention to those using bare lithium metal after 300 cycles. My comment concerns the prospective cyclic performance after further battery operation, since in figure 5c, the contrast of capacity retention between experiment and control group after 500 cycles seems more distinct. Consequently, the detailed comparison in context in terms of the 500 cycle would make the paper more impactful.

Reviewer #3 (Remarks to the Author):

The authors presented an interesting work, using a treated eggshell membrane to stabilize the lithium metal anode. They claimed that it is because of the strong affinity between the elements of the peptide chains and the lithium ions in the electrolytes that uniform lithium ion flux and thereafter uniform deposition of lithium metal was achieved. The manuscript was very well written. I have several concerns:

1. There have been quite a lot of papers reporting stable cycling of lithium metal anodes at the areal capacity around 1 mAh/cm², exploiting different strategies. Seldom can one find stable cycling at

areal capacities higher than this value (there are ones claiming stable cycling at $> 10 \text{ mAh/cm}^2$, but ironically using bulky ultraporous current collectors). However, to pair with practical commercial lithium-ion cathodes, the areal capacity should go higher than 3 mAh/cm^2 . Check for example the paper by Jiao et. al. Behavior of Lithium Metal Anodes under Various Capacity Utilization and High Current Density in Lithium Metal Batteries, published in Joule in 2018.

So the results presented in the current manuscript, while are solid and interesting, have only a very limited impact. The authors also did not provide the actual loading of their LiFePO_4 cathode for the readers to estimate how much lithium was deposited on the copper current collector, during the full cell experiments shown in Fig. 5.

2. The design of the experiment shown in Fig. 4a-f is fundamentally problematic. The half area covered by the TESM will naturally receive much fewer lithium ions to be deposited, due to the tortuous long path through the TESM, so lithium ions in the liquid electrolyte would have preferentially go to the other half without the $90\text{-}\mu\text{m}$ -thick porous TESM layer. The experiments should have been down in separate coin cells. Still, results for much higher areal capacity are much more meaningful.

3. The perceived nucleation mechanism is fundamentally flawed. Even though lithium ions may have a strong affinity with the TESM, they may not be utilized during the electrodeposition process, since the nucleation will occur beneath the solid-electrolyte interphase (SEI) layer, the process will only take up ions that have diffused across the SEI layer, not those still attracted by the TESM yet outside the SEI layer.

Especially when talking about the mechanism of nucleation and growth of lithium metal, the process is complicated by the SEI formation, the lithium structures the authors observed in their experiments, although very thin and long, are not dendrites, but lithium whiskers, which grow from the root and were pushed out from below the SEI layer. Check the paper by Bai et. al. Transition of lithium growth mechanisms in liquid electrolytes, published in Energy & Environmental Science in 2016. Without discussing the role of SEI, it's invalid to establish any correlation between lithium ions stored in the electrolyte-filled pore walls and the metal nucleation behaviors below the SEI layers.

In summary, the present study failed to consider the key issues of lithium metal anodes listed above. I cannot make a recommendation for acceptance.

Response to Reviewer 1:

Comment 1: In this work to improve battery performance, Tao and colleagues fabricate a membrane film composed of an eggshell protein (isolated from commercial eggshells) that is further chemically modified by a solvothermal process with trifluoroethanol. This porous membrane film termed "TESM" is then used as a coating on either a Cu or metallic Li foil substrate to influence the nucleation and growth of Li metal. The authors propose that in addition to homogenizing the distribution of Li ions near the substrate, the peptide chains in the TESM show an increased binding energy to (111) plane of Li metal according to DFT calculations. Both theoretical (DFT) and experimental (e.g. cryo-EM) techniques are used to support this claim. Improved electrochemical performance of both full cells and half cells are demonstrated with the TESM coating strategy.

Generally, it is quite interesting and striking that the normally dendritic Li metal grows as a spherical morphology with the TESM film (assuming this effect is in carbonate-based electrolyte). The only other work this reviewer is aware of that shows this effect in carbonate electrolytes is (Nature communications 9.1 (2018): 3656.). However, this manuscript lacks detail regarding the experimental procedures and protocols necessary to fully judge both the proposed mechanism and the electrochemical performance. If the mechanism of how Li metal changes from a dendritic to spherical morphology can be explained, this work would be of interest to the battery community. Unfortunately, this is not the case for the manuscript in its current form and this reviewer cannot support publication of this work in Nature Communications until the issues below are appropriately addressed.

1. The cryo-EM images in Fig. 3 are indeed beautifully taken, showing the crystalline Li metal along various zone axes. However, no detail is provided as to which electrolyte the Li metal was formed in. Upon first glance, Fig. 3a and 3d appear to be grown in DOL/DME electrolyte with 1% nitrate, which is widely known to nucleate Li metal in a spherical morphology. Fig. 3g-i then appear to be grown in standard carbonate-based electrolytes. Please provide details here. This reviewer will assume the electrolyte used in Fig. 3a,d and 3g-i are both carbonate-based electrolytes, otherwise there is no novelty as ether electrolytes with nitrate additives are known to influence the Li metal morphology.

Response: Thank you for the comment. Actually, Fig. 3a, 3d and 3g-i all exhibit the Li grown in ether electrolytes. Here, we further check the Li growth behavior in carbonate-based electrolytes. We have added this part on Page 7-Line 21 to demonstrate the sufficient novelty.

"Particularly, the Li deposition in standard carbonate-based electrolytes was also investigated, aiming at the wide application in future.⁵⁶ Similarly, the Li nucleate with the uniform spherical morphology on the TESM protected Cu while those deposits on the bare Cu tend to be dendritic (Supplementary Fig. 10)."

Supplementary Figure 10. SEM and Cryo-TEM images of Li deposits (a, c) with and (b, d) without TESM at a current density of 0.5 mA cm^{-2} with a capacity of 0.1 mAh cm^{-2} in carbonate-based electrolytes. Scale bars, (a, b) $10 \text{ }\mu\text{m}$, (c) $5 \text{ }\mu\text{m}$ and (d) $2 \text{ }\mu\text{m}$.

Comment 2: 2. The authors claim that with the TESM membrane, dendrite growth is suppressed due to a combination of providing a uniform distribution of Li ions and a direct influence on Li metal nucleation. There may indeed be an effect of the chemical groups of the TESM to help reduce the concentration gradient of Li ions. However, the explanation regarding the interaction between the peptide chains of the TESM and various crystalline facets of the Li metal does not make sense. Authors state that "binding energy of peptide chains on Li (110) and (211) slabs are lower than that on Li (111) slab" (line 160-163), which inhibits "growth of Li deposits along <111> direction" (line 169). If this were the case, Li metal should not grow as spherical structures and instead continue to grow as dendritic structures, just along the <211> or <110> directions. In order to grow as a spherical morphology, the Li metal would need to grow isotropically (by definition) with no preference towards any crystalline facet. However, the DFT calculations by the authors reveal otherwise. Furthermore, the calculations fail to consider the effect of the solid electrolyte interphase (SEI), which prevents direct contact between the Li metal surface and the peptide chains in the TESM. It is possible that chemical species in the TESM are

soluble in electrolyte which would then directly influence the Li metal nucleation. This should then be studied using cryo-EM imaging and spectroscopy to see if such species are incorporated into the SEI.

Response: Thank you for the comment. We have further calculated the binding energy of peptide chains on the most expressed slabs with low index (100), (210) and (310). The results indicate that the dendritic growth slabs, particularly the (111), are prone to be inhibited by the peptide chains. Meanwhile, the binding energy among the varied slabs have little difference, suggesting a isotropical growth without preferred crystalline orientation. The detailed discussion is presented on Page 7-Line 24.

“To figure out the impact of TESP on the growth direction of Li deposits, the binding energy of peptide chains on different planes of bulk Li metal were thereupon simulated by DFT calculations. We choose the most expressed Li slabs with low index (100), (210), (310) together with the slabs (110), (211), (111) towards which the dendrites grow, to calculate the binding energy coupling peptide chains. The results depict that the binding energy of peptide chains on Li (110) and (211) slabs are dramatically lower than that on Li (111) slab no matter without or with lithiation upon the peptides (Fig. 3j-l and Supplementary Fig. 11). Besides, the calculated differential charge density illustrates significant charge transfer between Li and peptide chain, especially for the case of the peptide chain adsorbed on Li (111) slab (Fig. 3j-l and Supplementary Fig. 11). Meanwhile, the binding energies among Li (100), (210), (310) slabs have little difference but exhibit a tendency lower than that on Li (110) and (211) slabs. These results reveal that the peptide chains have stronger inhibition effect on the growth of Li dendrites, particularly along <111> direction, by adsorbing on the relevant specific slab, reconfirming the observations in cryo-TEM.”

Supplementary Figure 11. **a-c** Representative snapshots of density functional theory-derived structures as well as differential charge density of lithiated peptide chain on Li **(a)** (110), **(b)** (211) and **(c)** (111) slabs. **d-i** Representative snapshots of density functional theory-derived structures as well as differential charge density of peptide chain **(d-f)** without and **(g-i)** with lithiation on Li **(d, g)** (100), **(e, h)** (210) and **(f, i)** (310) slabs.

Further, we investigate the composition of SEI using cryo-TEM imaging and spectroscopy (EDX, XPS, BCA and FTIR) in the light of your suggestion. According to a series of experimental results, the protein that is the typical species of TESP is certificated to be incorporated inside the SEI as you predicted. As such, the Li nucleation is regulated by the TESP involved SEI. The detailed discussion is added on Page 8-Line 10 and Page 15-Line 21.

“Furthermore, we proceed to investigate how the SEI influences the nucleation and growth of Li, which is essentially important to reveal the Li deposition behavior.⁵⁷⁻⁵⁹ According to the quantitative elemental analysis under cryo-TEM and the X-ray photoelectron spectroscopy (XPS) tests, the contents of N and S in Li deposited on TESP/Cu increased dramatically compared to that on the bare Cu (Supplementary Fig. 12-14). These results indicate that the TESP might participate in the formation of SEI. To verify the existence of the TESP in SEI, the bicinchoninic acid (BCA) assay was employed to detect the content of protein, the typical chemical species in TESP, in the SEI of both TESP/Li and bare Li. BCA assay explicitly demonstrated that the SEI of TESP/Li involved the protein of TESP while the one of bare Li did

not (Supplementary Fig. 15), as reconfirmed by the FTIR analysis (Supplementary Fig. 16). The experiments above suggest that part of the protein, the chemical species in the TSM are soluble in electrolyte which could be incorporated inside the SEI. The TSM macromolecules embedded SEI might contribute to adjust the crystallization behavior of Li, resulting in the sphere morphology of Li deposition.”

“Besides, the BCA protein assay and FTIR analysis confirm that part of the soluble protein species in TSM contribute to construct the SEI (Supplementary Fig. 15, 16). According to the recent report, the additive of NO_3^- could generate low-resistance SEI and thus contribute to the Li deposits on bulk surface, instead of the preferential dendritic deposition on tips/kinks.⁵⁶ Hence, to improve the migration of lithium ions across SEI is crucial in generating uniform spherical Li deposits. Similarly, in this work, the cell with TSM/Li exhibits the dramatically lower resistance compared to that with bare Li (Supplementary Fig. 22). As such, the protein-based macromolecules embedded SEI might induce the nucleation and growth of Li spheres by reducing the resistance, declaring another possible reason for suppressing Li dendrites.”

Supplementary Figure 12. **a** Cryo-STEM image of the Li deposited without TSM in ether-based electrolytes at 1 mA cm^{-2} with a capacity of 0.5 mAh cm^{-2} . **b** The corresponding elemental mapping images of the area highlighted in **(a)**. Scale bars, **(a)** 500 nm and **(b)** 200 nm. **c** The spectrum of element intensity obtained from **(b)**. **d** The relevant atom and mass fraction of varied elements measured in **(b)**.

Supplementary Figure 13. **a** Cryo-STEM image of the Li sphere formed in presence of TESM in ether-based electrolytes at 1 mA cm^{-2} with a capacity of 0.5 mAh cm^{-2} **b** The corresponding elemental mapping images of the area highlighted in **(a)**. Scale bars, **(a)** 500 nm and **(b)** 200 nm. **c** The spectrum of element intensity obtained from **(b)**. **d** The relevant atom and mass fraction of varied elements measured in **(b)**.

Supplementary Figure 14. XPS characterizations of Li foil coated without and with TESM after 20 cycles at 2 mA cm^{-2} with a capacity of 1 mAh cm^{-2} .

Supplementary Figure 15. a The BCA assay showing the typical chromogenic reaction of standard proteins, SEI on TESM/Li, and SEI on bare Li in BCA reagent. The color of standard sample changes from pale green to deep purple as the protein concentration increases. **b** The curve of the absorbance at 562 nm vs protein concentration plotted by the standard protein sample in (a), where the protein concentrations of the SEI on TESM/Li or bare Li are highlighted (The protein concentration of SEI on TESM/Li is 0.207 $\mu\text{g}/\mu\text{L}$ and the SEI on bare Li did not contain protein within the error range).

Supplementary Figure 16. FTIR spectra of the TESM, SEI on TESM/Li, and SEI on bare Li after 20 cycles at 2 mA cm^{-2} with 1 mAh cm^{-2} , where the typical bands of amide I, II, III are highlighted.

Comment 3: 3. In order to fairly judge the electrochemical performance of both the control and sample of interest, authors need to provide detailed information on battery components. For example, the electrolyte volume and N:P ratio, which greatly impact battery performance (Nature nanotechnology 14.6 (2019): 594.), need to be explicitly stated for each set of electrochemical data presented in the manuscript and supplemental materials.

Response: Thank you for the comment. We have checked the electrolyte volume and the N:P ratio, and added the detailed illustration in method part on Page 18-Line 17:

“The electrolyte volume used in this work is $60 \mu\text{L}$, the capacity ratio of the negative electrode to the positive electrode (N:P ratio)²⁰ is about 70 (the Li anode employed in this work is abundant and they are consistent in all the experiments), and the loading of LiFePO_4 cathode is ranging from 2.3 to 5.2 mg cm^{-2} .”

Comment 4: 4. How was the TESM placed onto the Cu or Li metal substrate? Does it serve as the separator or was the TESM a coating? Please provide details in either main text or supplemental materials.

Response: Thank you for the comment. TESM was directly coated on the Li substrate to serve as an interfacial layer. We have added this part on Page 6-Line 11:

“A dendrite-free Li morphology with spherical microstructure can be observed in the presence of TESM (The TESM was coated on Li to serve as an interfacial layer), whilst long Li dendrites are everywhere on the bare Cu without the protection of TESM (Fig. 2e, f).”

Response to Reviewer 2:

Comment 1: This manuscript describes a facile approach of natural biomolecules utilization for generating dendrite-free lithium metal anodes. This bio-inspired design proposes a relatively unique experiment, but cleverly performed and with interesting results: the organic matrix derived from trifluoroethanol-modified eggshell membrane significantly reduces the oriented growth of lithium dendrites along <111>, achieving the lithium metal-based rechargeable battery with significantly boosting cyclic performance. What impresses me is the atomic-resolution imaging through cryo-transmission electron microscopy, in visualizing the origins of lithium dendrites killing. The clear exhibition of the lithium atoms alignments coupled with computational simulation confirm that the remarkable adsorptions between biomolecule groups and specific lithium crystal faces are responsible for dendritic preferred <111> inhibition. The mechanism above is convincing and consistent well with how lives construct hierarchical crystalline structure through biomineralization. The results presented by authors have motivated a novel insight for deeply understanding the troublesome lithium dendrite growth, regarding controlling the lithium crystallization, which is even ingeniously inspired and learned from nature.

In conclusion, I like the story, and I recommend the publication of this manuscript in Nature Communications. Before the acceptance, I have however some comments that require explanations:

1. In Line 5, Page 6, the statement “Therefore, with the enhanced mechanical property and lithium-ion conductivity, the TESM may be more favorable for homogenizing the distribution of lithium ions, reducing inner resistance of cells, and inhibiting the dendritic Li growth.”, in my opinion, is not accuracy enough considering the causality of results and conclusions. To inhibit the lithium dendrites through homogenizing the lithium ions and thus reducing the local lithium ions flux should be attributed to the reason of both the uniform porous structure and the strong chemical affinity between lithium ions and organic matrix. The authors should rephrase the discussion of this part at the proper position in manuscript.

Response: Thank you for your suggestion. We agree with your opinion and we have revised the discussion of this part on Page 6-Line 24:

“Therefore, with the uniform porous structure and the strong chemical affinity between lithium ions and organic matrix, the TESM may be more favorable for homogenizing the distribution of lithium ions, reducing inner resistance of cells, and inhibiting the dendritic Li growth.^{55”}

Comment 2: 2. In DFT calculations, why do the authors construct two configurations, i.e., peptide chain with or without lithiation on Li slab, and what is the difference between these two situations since the results almost remain unchanged. This hypothesis is crucial to the conclusion so the authors should offer us more detailed discussion.

Response: Thank you for your suggestion. We believe that part of the TESP would be lithiated, particularly to participate the formation of SEI. In order to comprehensively understand how TESP influence the crystallization behavior of Li, we constructed two configurations of peptide chain with or without lithiation on Li slab consequently. Further experiments confirmed that the components of TESP indeed existed in the SEI layer. The detailed discussions are presented on Page 8-Line 10 and Page 15-Line 21:

“Furthermore, we proceed to investigate how the SEI influences the nucleation and growth of Li, which is essentially important to reveal the Li deposition behavior.⁵⁷⁻⁵⁹ According to the quantitative elemental analysis under cryo-TEM and the X-ray photoelectron spectroscopy (XPS) tests, the contents of N and S in Li deposited on TESP/Cu increased dramatically compared to that on the bare Cu (Supplementary Fig. 12-14). These results indicate that the TESP might participate in the formation of SEI. To verify the existence of the TESP in SEI, the bicinchoninic acid (BCA) assay was employed to detect the content of protein, the typical chemical species in TESP, in the SEI of both TESP/Li and bare Li. BCA assay explicitly demonstrated that the SEI of TESP/Li involved the protein of TESP while the one of bare Li did not (Supplementary Fig. 15), as reconfirmed by the FTIR analysis (Supplementary Fig. 16). The experiments above suggest that part of the protein, the chemical species in the TESP are soluble in electrolyte which could be incorporated inside the SEI. The TESP macromolecules embedded SEI might contribute to adjust the crystallization behavior of Li, resulting in the sphere morphology of Li deposition.”

“Besides, the BCA protein assay and FTIR analysis confirm that part of the soluble protein species in TESP contribute to construct the SEI (Supplementary Fig. 15, 16). According to the recent report, the additive of NO_3^- could generate low-resistance SEI and thus contribute to the Li deposits on bulk surface, instead of the preferential dendritic deposition on tips/kinks.⁵⁶ Hence, to improve the migration of lithium ions across SEI is crucial in generating uniform spherical Li deposits. Similarly, in this work, the cell with TESP/Li exhibits the dramatically lower resistance compared to that with bare Li (Supplementary Fig. 22). As such, the protein-based macromolecules embedded SEI might induce the nucleation and growth of Li spheres by reducing the resistance, declaring another possible reason for suppressing Li dendrites.”

Comment 3: 3. The enhanced cycling capabilities of full cells using TESP / lithium metal are demonstrated by making a comparison of capacity retention to those using bare lithium metal after 300 cycles. My comment concerns the prospective cyclic performance after further battery operation, since in figure 5c, the contrast of capacity retention between experiment and control group after 500 cycles seems more distinct. Consequently, the detailed comparison in context in terms of the 500 cycle would make the paper more impactful.

Response: Thank you for your suggestion. We compared the capacity retention between experiment and control group after 500 cycles as you suggested. We have revised this part on Page 13-Line 12:

“The capacity retention is still above 85 % and cells with TESM/Li show prolonged lifespan more than 500 cycles. In contrast, cells with bare Li foil appear a fast decay in capacity and cycling (below 60 % capacity retention after 500 cycles).”

Fig. 5 Electrochemical performance of bare Li, ESM/Li, or TESM/Li electrodes. **a-d** Voltage profiles of symmetric cells at **(a)** 1 mA cm⁻², **(b, c)** 5 mA cm⁻² or **(d)** 10 mA cm⁻² with a Li capacity of **(a, b)** 1 mAh cm⁻², **(c)** 3 mAh cm⁻² or **(d)** 5 mAh cm⁻². **e** Long-term cycling lifespan of Li/LiFePO₄ full cells at 1 C. **f** Rate performance from 0.2 to 5 C.

Response to Reviewer 3:

Comment 1: The authors presented an interesting work, using a treated eggshell membrane to stabilize the lithium metal anode. They claimed that it is because of the strong affinity between the elements of the peptide chains and the lithium ions in the electrolytes that uniform lithium ion flux and thereafter uniform deposition of lithium metal was achieved. The manuscript was very well written. I have several concerns:

1. There have been quite a lot of papers reporting stable cycling of lithium metal anodes at the areal capacity around 1 mAh/cm², exploiting different strategies. Seldom can one find stable cycling at areal capacities higher than this value (there are ones claiming stable cycling at > 10 mAh/cm², but ironically using bulky ultraporous current collectors). However, to pair with practical commercial lithium-ion cathodes, the areal capacity should go higher than 3 mAh/cm². Check for example the paper by Jiao et. al. Behavior of Lithium Metal Anodes under Various Capacity Utilization and High Current Density in Lithium Metal Batteries, published in Joule in 2018.

So the results presented in the current manuscript, while are solid and interesting, have only a very limited impact. The authors also did not provide the actual loading of their LiFePO₄ cathode for the readers to estimate how much lithium was deposited on the copper current collector, during the full cell experiments shown in Fig. 5.

Response: Thank you for the comment. We have conducted the further experiments to find the stable cycling performance of the modified lithium metal anodes with the areal capacity higher than 3 mAh cm⁻² as you suggested. The detailed description is presented on Page 13-Line 1 and the updated Fig. 5.

“Considering the request of the wide application in pairing with lithium-ion cathodes, we then improved the areal capacity higher than 3 mAh cm⁻².⁶¹ The TESM/Li anodes under 3 mAh cm⁻² and 5 mAh cm⁻² both show stable cycling performance over 600 h with lower overpotential of ~ 15 mV and ~ 40 mV, respectively (Fig. 5c, d).”

Meanwhile, the actual loading of the LiFePO₄ is ranging from 2.3 to 5.2 mg cm⁻², which is added in the methods part. Particularly, even if the loading of LiFePO₄ is further increased, the TESM-Li still exhibits stable cycling performance according to our current experiments.

Fig. 5 Electrochemical performance of bare Li, ESM/Li, or TESM/Li electrodes. **a-d** Voltage profiles of symmetric cells at **(a)** 1 mA cm⁻², **(b, c)** 5 mA cm⁻² or **(d)** 10 mA cm⁻² with a Li capacity of **(a, b)** 1 mAh cm⁻², **(c)** 3 mAh cm⁻² or **(d)** 5 mAh cm⁻². **e** Long-term cycling lifespan of Li/LiFePO₄ full cells at 1 C. **f** Rate performance from 0.2 to 5 C.

Comment 2: 2. The design of the experiment shown in Fig. 4a-f is fundamentally problematic. The half area covered by the TESM will naturally receive much fewer lithium ions to be deposited, due to the tortuous long path through the TESM, so lithium ions in the liquid electrolyte would have preferentially go to the other half without the 90-um-thick porous TESM layer. The experiments should have been done in separate coin cells. Still, results for much higher areal capacity are much more meaningful.

Response: Thank you for the comment. We have changed the experiments in Fig. 4a-f into the tests of separate coin cells as you suggested. Moreover, the results for much higher areal capacity are also presented. The detailed discussion of this part is on Page 10-Line 2:

“Both the Li covered with TESM and the bare Li were employed to investigate the difference upon cycling. The TESM-protected coin cell presents silvery white (Fig. 4a) while the corresponding unprotected one goes black (Fig. 4d). Specifically speaking, there are pronounced mossy Li formed on the unprotected coin cell (Fig. 4e, f). In contrast, the TESM-protected Li anode has a fairly smooth and dense surface without the formation of dendritic or dead Li (Fig. 4b, c). Even if the areal capacity increases to 3 mAh cm^{-2} , the TESM/Li still possesses rather uniform morphology compared with the bare Li after the cyclic operation (Supplementary Fig. 17).”

Fig. 4 Morphology and cycling stability of Li anode assisted by TESM. **a, d** Optical photographs of (a) Li foil protected by TESM and (d) bare Li foil after 20 cycles at 2 mA cm^{-2} for 1 mAh cm^{-2} . **b, c, e, f** SEM images and corresponding enlarged areas of (b, c) the TESM-protected Li and (e, f) the unprotected Li, respectively. Scale bars, (b, e) $50 \mu\text{m}$, and

(c, f) 20 μm . g Comparison of CEs of cells with bare Cu, ESM-modified Cu, and TESM-modified Cu at various current densities. The fixed cycling capacity in each cycle is 1 mAh cm^{-2} . h The voltage profiles for electrodes at 3 mA cm^{-2} . i The corresponding voltage hysteresis of electrodes with the current density of 3 mA cm^{-2} .

Supplementary Figure 17. a, b SEM images of (a) the TESM-protected Li anode and (b) the unprotected Li anode at 5 mA cm^{-2} with a Li capacity of 3 mAh cm^{-2} after 20 cycles, respectively. Scale bars, (a, b) 50 μm .

Comment 3: 3. The perceived nucleation mechanism is fundamentally flawed. Even though lithium ions may have a strong affinity with the TESM, they may not be utilized during the electrodeposition process, since the nucleation will occur beneath the solid-electrolyte interphase (SEI) layer, the process will only take up ions that have diffused across the SEI layer, not those still attracted by the TESM yet outside the SEI layer.

Especially when talking about the mechanism of nucleation and growth of lithium metal, the process is complicated by the SEI formation, the lithium structures the authors observed in their experiments, although very thin and long, are not dendrites, but lithium whiskers, which grow from the root and were pushed out from below the SEI layer. Check the paper by Bai et. al. Transition of lithium growth mechanisms in liquid electrolytes, published in Energy & Environmental Science in 2016. Without discussing the role of SEI, it's invalid to establish any correlation between lithium ions stored in the electrolyte-filled pore walls and the metal nucleation behaviors below the SEI layers.

In summary, the present study failed to consider the key issues of lithium metal anodes listed above. I cannot make a recommendation for acceptance.

Response: Thank you for the comment. We agree that the SEI is essentially crucial in influencing the Li nucleation behaviors. Hence, we checked the composition of the SEI through further experiments. It is found that the chemical species of TESM are incorporated inside the SEI, thus regulating the nucleation and growth of the Li. We have added the detailed discussion on Page 8-Line 10 and Page 15-Line 21.

Furthermore, we proceed to investigate how the SEI influences the nucleation and growth of Li, which is essentially important to reveal the Li deposition behavior.⁵⁷⁻⁵⁹

According to the quantitative elemental analysis under cryo-TEM and the X-ray photoelectron spectroscopy (XPS) tests, the contents of N and S in Li deposited on TESM/Cu increased dramatically compared to that on the bare Cu (Supplementary Fig. 12-14). These results indicate that the TESM might participate in the formation of SEI. To verify the existence of the TESM in SEI, the bicinechonic acid (BCA) assay was employed to detect the content of protein, the typical chemical species in TESM, in the SEI of both TESM/Li and bare Li. BCA assay explicitly demonstrated that the SEI of TESM/Li involved the protein of TESM while the one of bare Li did not (Supplementary Fig. 15), as reconfirmed by the FTIR analysis (Supplementary Fig. 16). The experiments above suggest that part of the protein, the chemical species in the TESM are soluble in electrolyte which could be incorporated inside the SEI. According to the recent report, the additive of NO_3^- could generate low-resistance SEI and thus contribute to the Li deposits on bulk surface, instead of the preferential dendritic deposition on tips/kinks.⁵⁶ Hence, to improve the migration of lithium ions across SEI is crucial in generating uniform spherical Li deposits. Similarly, in this work, the cell with TESM/Li exhibits the dramatically lower resistance compared to that with bare Li (Supplementary Fig. 22). As such, the protein-based macromolecules embedded SEI might induce the nucleation and growth of Li spheres by reducing the resistance, declaring another possible reason for suppressing Li dendrites.

Meanwhile, we have carefully checked the images of Figure 3g-i to examine the Li structures. We agree that the Li deposits imaged in Figure 3g-i might be Li whiskers according to the research: Bai, P. et al. *Energy Environ. Sci.* **9**, 3221 (2016); Bai, P. et al. *Joule* **2**, 2434 (2018). It is noticed that, the morphology and size of Li whiskers and Li dendrites vary greatly in different electrolytes and current density. The ribbon-like Li we imaged here possessed ~ 350 nm diameters. Particularly, some groups observed the dendritic Li through TEM which is quite similar with what we imaged in this work (Extended Figure). They termed these Li deposits with dendritic feature as “Li dendrites” in paper: Li, Y. et al. *Science* **358**, 506 (2017); Hong, D. et al. Li, Y. et al. *Joule* **2**, 2167 (2018); *J. Mater. Chem. A* **7**, 20325 (2019). As such, we believe it is really hard to distinguish the Li whiskers and Li dendrites through the TEM observation. Nevertheless, the Li whiskers and Li dendrites are likely to nucleate or grow through the consistent orientation as references discussed: He, Y. et al. *Nat. nanotechnol.* DOI: 10.1038/s41565-019-0558-z (2019); Cheng, X. B., Zhang, R., Zhao, C. Z. & Zhang, Q. *Chem. Rev.* **117**, 10403 (2017). As such, the crystallographic orientations of dendritic Li we demonstrated in Figure 3g-i are convincible, which greatly helps figure out how TESM influence the Li deposition at the atom-scale.

We feel grateful to your constructive comments. We further consider the key issues of Li anodes as you suggested. In the revised manuscript, we have carefully conducted further experiments and characterizations, as well as discussion for the investigations of the cycling stability and morphology change of Li anodes under high areal capacities. Particularly, the fundamental understanding of how SEI influences the nucleation and growth of Li is clarified. All the above questions raised by the reviewer have been addressed and the manuscript has been greatly improved now. We sincerely hope the reviewer could reconsider and revalue the impact of our work after this revision.

Supplementary Figure 12. **a** Cryo-STEM image of the Li deposited without TESM in ether-based electrolytes at 1 mA cm^{-2} with a capacity of 0.5 mAh cm^{-2} . **b** The corresponding elemental mapping images of the area highlighted in **(a)**. Scale bars, **(a)** 500 nm and **(b)** 200 nm. **c** The spectrum of element intensity obtained from **(b)**. **d** The relevant atom and mass fraction of varied elements measured in **(b)**.

Supplementary Figure 13. **a** Cryo-STEM image of the Li sphere formed in presence of TESM in ether-based electrolytes at 1 mA cm^{-2} with a capacity of 0.5 mAh cm^{-2} **b** the corresponding elemental mapping images of the area highlighted in **(a)**. Scale bars, **(a)** 500 nm and **(b)** 200 nm. **c** The spectrum of element intensity obtained from **(b)**. **d** The relevant atom and mass fraction of varied elements measured in **(b)**.

Supplementary Figure 14. XPS characterizations of Li foil coated without and with TESM after 20 cycles at 2 mA cm^{-2} with a capacity of 1 mAh cm^{-2} .

Supplementary Figure 15. a The BCA assay showing the typical chromogenic reaction of standard proteins, SEI on TESM/Li, and SEI on bare Li in BCA reagent. The color of standard sample changes from pale green to deep purple as the protein concentration increases. **b** The curve of the absorbance at 562 nm vs protein concentration plotted by the standard protein sample in (a), where the protein concentrations of the SEI on TESM/Li or bare Li are highlighted (The protein concentration of SEI on TESM/Li is 0.207 $\mu\text{g}/\mu\text{L}$ and the SEI on bare Li did not contain protein within the error range).

Supplementary Figure 16. FTIR spectra of the TESM, SEI on TESM/Li, and SEI on bare Li after 20 cycles at 2 mA cm^{-2} with 1 mAh cm^{-2} , where the typical bands of amide I, II, III are highlighted.

[Redacted]

Extended Figure. TEM images of ribbon-like Li with the diameters ranging from 300-450 nm. Images in this work are presented in first row and images in referenced work (Li, Y. et al. *Science* 358, 506-510 (2017); Hong, D. et al. Li, Y. et al. *Joule* 2, 2167-2177 (2018); *J. Mater. Chem. A* 7, 20325-20334 (2019).) are presented in second row.

Reviewers' comments:

Reviewer #1 (Remarks to the Author):

Review attached

In evaluating the authors' response and overall work, this reviewer examines both the (1) electrochemical performance and the (2) fundamental mechanism proposed by the authors.

Electrochemical Performance:

In my first review, details about battery fabrication were lacking to fairly judge the electrochemical performance. With the additional details provided (e.g. N:P ratio of 70, electrolyte volume of 60 μL , 2.3 – 5.2 mg LFP loading), this reviewer finds the current electrochemical performance with the TESM coating strategy somewhat lackluster. Firstly, the authors describe a range of 2.3 -5.2 mg LFP loading, but fail to state what the exact loading of each curve is in Fig. 5e-f. If the loading is different, this could easily explain discrepancy between the different conditions. Even assuming the ideal condition that every curve is the highest loading (5.2 mg LFP), this only provides a maximum capacity of $\sim 0.6 \text{ mAh/cm}^2$, which is significantly below the industrial level of 3.0 mAh/cm^2 . Furthermore, the authors' choice to switch between carbonate-based electrolytes and ether-based electrolytes in both characterization and electrochemical testing experiments makes it difficult for this reviewer to compare between each condition. Why not just do all the testing in one standard electrolyte? Although the electrochemical performance shows improvement with the TESM coating strategy, this reviewer suggests doing a higher loading of LFP with either a lower NP ratio using thin Li metal foil or anode-free Cu current collector. This reviewer understands that these experiments are time-consuming and overall challenging, but previous battery research in *Nature Communications* have shown such experiments.

Fundamental Understanding:

For the fundamental mechanism proposed by the authors of how the TESM enhances uniform Li nucleation and growth, this reviewer finds the DFT results unconvincing. The DFT results on binding energy between different facets of Li metal and the peptide chain to explain isotropic growth of Li metal still does not make sense. Authors show the binding energy for the "most expressed" slabs (100), (210), and (310) are lower than the (110), (211), and (111) slabs. So shouldn't this mean that now the (100), (210), and (310) facets are more favorable for the addition of new Li atoms? This does not explain the isotropic growth observed in the electron micrographs. In this reviewer's opinion, the DFT results are unconvincing since the model used to calculate energies does not reflect the real condition, which is Li metal interfaced with the SEI layer. Fresh Li metal will not be interfaced with the peptide chain in the TESM coating under deposition or stripping conditions. However, the authors do provide new data showing the chemistry of the SEI with and without TESM coating. The elemental mapping and assay do reveal components of the TESM coating may be incorporated into the SEI, which could influence the transport of Li ions. This should be further explored to bolster the authors' proposed mechanism. For example, how does the TESM get incorporated into the SEI? Is the TESM incorporated into the SEI purely from being soluble in the electrolyte? Or does the TESM first decompose (before electrolyte decomposition) to form a passivation layer onto the Cu surface or Li surface, which may help provide a uniform Li flux for deposition? What then is the nature of this layer? Cryo-EM can help probe the structure and chemistry, while CV would be able to elucidate at which voltage the TESM decomposes (if indeed it does). These are examples of the questions the authors should pursue to further understand their operating mechanism.

Verdict:

Based on the above comments, this reviewer finds the current electrochemical performance lackluster, but could be improved with further experiments. The proposed mechanism to explain the Li deposition behavior is incomplete. This reviewer is unconvinced by the DFT results and hypothesize that other processes may be at play. The biggest hint is the chemical signature of the TESM components in the SEI. In summary, this reviewer would be open to seeing a revised manuscript that addresses these shortcomings and others brought up by the other reviewers.

Reviewer #2 (Remarks to the Author):

The revision is satisfactory and the manuscript is recommended for publication.

Reviewer #3 (Remarks to the Author):

I appreciate the efforts and new results the authors put together for the revision, which indeed successfully addressed my earlier concerns. I believe it is now in a great shape to be accepted by Nature Communications.

Response to Reviewer 1:

Comment 1: In evaluating the authors' response and overall work, this reviewer examines both the (1) electrochemical performance and the (2) fundamental mechanism proposed by the authors.

Electrochemical Performance: In my first review, details about battery fabrication were lacking to fairly judge the electrochemical performance. With the additional details provided (e.g. N:P ratio of 70, electrolyte volume of 60 μL , 2.3 – 5.2 mg LFP loading), this reviewer finds the current electrochemical performance with the TESM coating strategy somewhat lackluster. Firstly, the authors describe a range of 2.3 - 5.2 mg LFP loading, but fail to state what the exact loading of each curve is in Fig. 5e-f. If the loading is different, this could easily explain discrepancy between the different conditions. Even assuming the ideal condition that every curve is the highest loading (5.2 mg LFP), this only provides a maximum capacity of $\sim 0.6 \text{ mAh/cm}^2$, which is significantly below the industrial level of 3.0 mAh/cm^2 .

Response: Thank you for the comment. In order to verify the electrochemical performance enhancement brought by this strategy, we have conducted the experiments with the LiFePO_4 cathode of about 3.0 mAh cm^{-2} ($\sim 22.4 \text{ mg cm}^{-2}$) under low N/P ratio (~ 3.3). We have added this part of discussion on Page 13-Line 7, Figure 5e and methods section in the revised manuscript:

“The cycling stability enhanced by the TESM was evaluated under high mass loading of LiFePO_4 ($\sim 3.0 \text{ mAh cm}^{-2}$) and low N/P ratio (~ 3.3) to meet the realistic condition. As shown in Fig. 5e, the cells with TESM/Li show prolonged lifespan more than 160 cycles and the capacity is still above 150 mAh g^{-1} with a steady CE of nearly 100 %. In sharp contrast, cells with bare Li appear a fast decay in capacity and cycling (below 50 % capacity retention after 20 cycles).”

“The electrolyte volume used in this work is 60 μL , the N/P ratio²⁰ is about 3.3 (the limited Li was deposited on Cu current collector with a fixed capacity of 10.0 mAh cm^{-2} at 1 mA cm^{-2} in ether-based electrolytes), and the mass loading of LiFePO_4 cathode is about 3.0 mAh cm^{-2} ($\sim 22.4 \text{ mg cm}^{-2}$). The diameter of the cathode and Cu current collector is 12 mm. All Li/ LiFePO_4 full cells were tested between 2.5 and 4.2 V at 1 C in ether-based electrolytes.”

Fig. 5 Electrochemical performance of bare Li, ESM/Li, or TESM/Li electrodes in ether-based electrolytes. **a-d** Voltage profiles of symmetric cells at **(a)** 1 mA cm⁻², **(b, c)** 5 mA cm⁻² or **(d)** 10 mA cm⁻² with a Li capacity of **(a, b)** 1 mAh cm⁻², **(c)** 3 mAh cm⁻² or **(d)** 5 mAh cm⁻². **e** Cycling performance of Li/LiFePO₄ full cells with the capacity ratio of the negative electrode to the positive electrode (N/P ratio) about 3.3 (Cu with limited Li: 10 mAh cm⁻², LiFePO₄: 3 mAh cm⁻²) at 1 C (1 C = 170 mA g⁻¹).

Comment 2: Furthermore, the authors' choice to switch between carbonate-based electrolytes and ether-based electrolytes in both characterization and electrochemical testing experiments makes it difficult for this reviewer to compare between each condition. Why not just do all the testing in one standard electrolyte?

Response: Thank you for the comment. In previous version, to address referee's concern about the Li morphology deposited in standard carbonate-based electrolyte, we visualized the Li deposition formed in carbonate-based electrolyte (Supplementary Fig. 10). In order to make the comparisons more apparent, in the current manuscript, we have replaced the full-cell test in carbonate-based electrolyte by that obtained in ether-based electrolyte with high LiFePO₄ loading and low N/P ratio (~ 3.3) in Fig. 5e. Hence, all the tests exhibited in main text now were conducted in ether-based standard electrolyte.

Comment 3: Although the electrochemical performance shows improvement with the TESM coating strategy, this reviewer suggests doing a higher loading of LFP with either a lower NP ratio using thin Li metal foil or anode-free Cu current collector. This reviewer understands that these experiments are time-consuming and overall challenging, but previous battery research in Nature Communications have shown such experiments.

Response: Thank you for the comment. We have conducted the further experiments of a higher mass loading of LiFePO₄ (about 3.0 mAh cm⁻²) with a lower N/P ratio (~ 3.3) using limited Li (10.0 mAh cm⁻²) as you suggested. The detailed exhibition is added on Page 13-Line 7, Figure 5e and methods section:

“The cycling stability enhanced by the TESM was evaluated under high mass loading of LiFePO₄ (~ 3.0 mAh cm⁻²) and low N/P ratio (~ 3.3) to meet the realistic condition. As shown in Fig. 5e, the cells with TESM/Li show prolonged lifespan more than 160 cycles and the capacity is still above 150 mAh g⁻¹ with a steady CE of nearly 100 %. In sharp contrast, cells with bare Li appear a fast decay in capacity and cycling (below 50 % capacity retention after 20 cycles).”

“The electrolyte volume used in this work is 60 μL, the N/P ratio²⁰ is about 3.3 (the limited Li was deposited on Cu current collector with a fixed capacity of 10.0 mAh cm⁻²), and the mass loading of LiFePO₄ cathode is about 3.0 mAh cm⁻² (~ 22.4 mg cm⁻²). The diameter of the cathode and Cu current collector is 12 mm. All Li/LiFePO₄ full cells were tested between 2.5 and 4.2 V at 1 C in ether-based electrolytes.”

Comment 4: Fundamental Understanding: For the fundamental mechanism proposed by the authors of how the TESM enhances uniform Li nucleation and growth, this reviewer finds the DFT results unconvincing. The DFT results on binding energy between different facets of Li metal and the peptide chain to explain isotropic growth of Li metal still does not make sense. Authors show the binding energy for the “most expressed” slabs (100), (210), and (310) are lower than the (110), (211), and (111) slabs. So shouldn't this mean that now the (100), (210), and (310) facets are more favorable for the addition of new Li atoms? This does not explain the isotropic growth observed in the electron micrographs. In this reviewer's opinion, the DFT results are unconvincing since the model used to calculate energies does not reflect the real condition, which is Li metal interfaced with the SEI layer. Fresh Li metal will not be interfaced with the peptide chain in the TESM coating under deposition or stripping conditions.

Response: Thank you for the comment. We agree that even if the TESM was embedded inside the SEI, the compositions and structures of SEI where Li interfaces the TESM are complicated. So it is an unbalanced view to attribute the reason of isotropic Li growth to the interaction between TESM and the specific crystal slabs. Consequently, we have deleted the part of DFT simulations of the binding energy in revised manuscript and added the further investigations of how TESM transform into the SEI, thus influencing the Li deposition instead.

Comment 5: However, the authors do provide new data showing the chemistry of the SEI with and without TESM coating. The elemental mapping and assay do reveal components of the TESM coating may be incorporated into the SEI, which could influence the transport of Li ions. This should be further explored to bolster the authors' proposed mechanism. For example, how does the TESM get incorporated into the SEI? Is the TESM incorporated into the SEI purely from being soluble in the electrolyte? Or does the TESM first decompose (before electrolyte decomposition) to form a passivation layer onto the Cu surface or Li surface, which may help provide a uniform Li flux for deposition? What then is the nature of this layer? Cryo-EM can help probe the structure and chemistry, while CV would be able to elucidate at which voltage the TESM decomposes (if indeed it does). These are examples of the questions the authors should pursue to further understand their operating mechanism.

Response: Thank you for your suggestion. We agree with your opinion and further explore the mechanism of TESM-incorporated-SEI enabling enhanced Li ions transport through the CV and Cryo-TEM analysis. The detailed discussion is added on Page 9-Line 1:

“Apart from the intrinsically soluble property of TESM in electrolyte proved above, the chemical nature of TESM during electrochemical process in terms of composing SEI was further investigated. As seen from the CV curves, the TESM exhibited a deep cathodic current peak around 1.4 V, while the bare Cu did not. (Supplementary Fig. 14). Thus, this reduction peak demonstrated that the TESM could be reduced

as the frontier reaction. Although the TESM was insulated by two-sided PP separator from the electrodes, the peaks remain the close-by position, reconfirming that the TESM was naturally soluble in electrolyte even without lithiation. Furthermore, the cryo-TEM visualization of the SEI constructed under different potentials help to monitor the composition change (Supplementary Fig. 15-20). It is noticed that the elemental ratio of N/C in TESM-present SEI increased significantly at -0.05 V (Supplementary Fig. 21). Considering the typical element N possessed in TESM, it is confirmed that more species of TESM would form the TESM-involved SEI, particularly at -0.05 V. The TESM macromolecules embedded SEI might contribute to provide a homogeneous lithium-ion flux, resulting in the sphere morphology of Li deposition.”

Supplementary Figure 14. Cyclic voltammetry of half cells corresponding to bare Cu, TESM, and TESM insulated by double-layer polypropylene (PP) at a scan rate of 1 mV s^{-1} in ether-based electrolyte.

Supplementary Figure 15. **a, b** Cryo-TEM and STEM image of the formed SEI at 1 V without TESM in ether-based electrolytes at 0.05 mA cm^{-2} . **c** The corresponding elemental mapping images in (b). Scale bars, (a-c) 200 nm. **d** The spectrum of element intensity obtained from (c). **e** The relevant atomic and mass fraction of varied elements measured in (c).

Supplementary Figure 16. **a, b** Cryo-TEM and STEM image of the formed SEI at 1 V in the presence of TESM in ether-based electrolytes at 0.05 mA cm^{-2} . **c** The corresponding elemental mapping images in (b). Scale bars, (a-c) 200 nm. **d** The spectrum of element intensity obtained from (c). **e** The relevant atomic and mass fraction of varied elements measured in (c).

Supplementary Figure 17. **a, b** Cryo-TEM and STEM image of the formed SEI at 0.01 V without TSM in ether-based electrolytes at 0.05 mA cm^{-2} . **c** The corresponding elemental mapping images in (b). Scale bars, (a-c) 200 nm. **d** The spectrum of element intensity obtained from (c). **e** The relevant atomic and mass fraction of varied elements measured in (c).

Supplementary Figure 18. **a, b** Cryo-TEM and STEM image of the formed SEI at 0.01 V in the presence of TESP in ether-based electrolytes at 0.05 mA cm^{-2} . **c** The corresponding elemental mapping images in **(b)**. Scale bars, **(a-c)** 200 nm. **d** The spectrum of element intensity obtained from **(c)**. **e** The relevant atomic and mass fraction of varied elements measured in **(c)**.

Supplementary Figure 21. The relevant mass ratio of elements N to C within SEI films formed without or with TESM under different potentials in ether-based electrolytes according to Supplementary Fig. 15-20.

Comment 6: Verdict: Based on the above comments, this reviewer finds the current electrochemical performance lackluster, but could be improved with further experiments. The proposed mechanism to explain the Li deposition behavior is incomplete. This reviewer is unconvinced by the DFT results and hypothesize that other processes may be at play. The biggest hint is the chemical signature of the TESM components in the SEI. In summary, this reviewer would be open to seeing a revised manuscript that addresses these shortcomings and others brought up by the other reviewers.

We feel grateful to your constructive comments. We further consider the key issues of the electrochemical performance together with the mechanism illustrating improved Li deposition as you suggested. In the revised manuscript, we have carefully conducted further experiments and characterizations, as well as discussion for the investigations of the cycling stability under the high mass loading of LiFePO_4 and low N/P ratio. Particularly, the in-depth understanding of how TESM participates in SEI and thus influences the growth of Li is further clarified. All the above questions raised by the reviewer have been addressed and the manuscript has been greatly improved now. We sincerely hope the reviewer could suggest the publication of our work after this revision.

REVIEWERS' COMMENTS:

Reviewer #1 (Remarks to the Author):

This reviewer appreciates the time and care the authors put forth in improving the manuscript. In my opinion, it is now ready for publication in Nature Communications.

Reviewer #1 (Remarks to the Author):

This reviewer appreciates the time and care the authors put forth in improving the manuscript. In my opinion, it is now ready for publication in Nature Communications.

Response to Reviewer 1:

Comment: This reviewer appreciates the time and care the authors put forth in improving the manuscript. In my opinion, it is now ready for publication in Nature Communications.

We feel grateful to your constructive comments for improving the quality and readability of manuscript.